# Origin and arrangement of actin filaments for gliding motility in apicomplexan parasites revealed by cryo-electron tomography

Matthew Martinez [1,7], Shrawan Kumar Mageswaran [1,2,7], Amandine Guérin[3], William David Chen[1], Cameron Parker Thompson[4], Sabine Chavin [5], Dominique Soldati-Favre [6], Boris Striepen [3] & Yi-Wei Chang [1,2,4] ✉

The phylum Apicomplexa comprises important eukaryotic parasites that invade host tissues and cells using a unique mechanism of gliding motility. Gliding is powered by actomyosin motors that translocate host-attached surface adhesins along the parasite cell body. Actin filaments (F-actin) generated by Formin1 play a central role in this critical parasitic activity. However, their subcellular origin, path and ultrastructural arrangement are poorly understood. Here we used cryo-electron tomography to image motile *Cryptosporidium parvum* sporozoites and reveal the cellular architecture of F-actin at nanometer-scale resolution. We demonstrate that F-actin nucleates at the apically positioned preconoidal rings and is channeled into the pellicular space between the parasite plasma membrane and the inner membrane complex in a conoid extrusion-dependent manner. Within the pellicular space, filaments on the inner membrane complex surface appear to guide the apico-basal flux of F-actin. F-actin concordantly accumulates at the basal end of the parasite. Finally, analyzing a Formin1-depleted *Toxoplasma gondii* mutant pinpoints the upper preconoidal ring as the conserved nucleation hub for F-actin in *Cryptosporidium* and *Toxoplasma*. Together, we provide an ultrastructural model for the life cycle of F-actin for apicomplexan gliding motility.

Some of the most important and widespread human parasites, including *Plasmodium, Toxoplasma*, and *Cryptosporidium*, which cause malaria, encephalitis, and severe diarrheal disease, respectively, belong to the eukaryotic phylum Apicomplexa[1–3]. These intracellular parasites have complex life cycles with distinct invasive stages that are adapted to different hosts and cell types. Although parasites in these stages are highly motile, they lack widely used motility organelles like flagella or pseudopodia. Instead, they rely on a unique form of locomotion known as helical gliding motility that is essential to navigate, invade, and egress from host cells[4–6]. According to a widely accepted

[1]Department of Biochemistry and Biophysics, Perelman School of Medicine, University of Pennsylvania, Philadelphia, PA, USA. [2]Institute of Structural Biology, Perelman School of Medicine, University of Pennsylvania, Philadelphia, PA, USA. [3]Department of Pathobiology, School of Veterinary Medicine, University of Pennsylvania, Philadelphia, PA, USA. [4]Pennsylvania Muscle Institute, Perelman School of Medicine, University of Pennsylvania, Philadelphia, PA, USA. [5]Department of Physics and Astronomy, University of Pennsylvania, Philadelphia, PA, USA. [6]Department of Microbiology and Molecular Medicine, Faculty of Medicine, University of Geneva, Geneva, Switzerland. [7]These authors contributed equally: Matthew Martinez, Shrawan Kumar Mageswaran. ✉e-mail: ywc@pennmedicine.upenn.edu

model, the parasite's myosin motors translocate filamentous actin (F-actin) rearward to power this mode of motility; the "apico-basal" flux of F-actin is converted to forward motion by linking the F-actin to parasite surface adhesins, which in turn interact with the extracellular matrix[5,7,8]. Much of our understanding of the nature of apicomplexan F-actin related to gliding motility comes from in vitro studies, mainly from *Plasmodium* and *Toxoplasma*[8]. Although there is an ongoing debate over the critical concentration and the ability to form long filaments[9–11], several studies indicate that apicomplexan F-actin is more unstable compared to its metazoan counterpart[12,13] and likely forms short filaments in vivo[14,15], a property that has likely hindered experimentation and conceptual advancement. Recently, fluorescence microscopy of host-penetrating *Toxoplasma* and *Plasmodium* cells has demonstrated the apico-basal flux of F-actin and its accumulation at the basal end[16,17]. However, how F-actin is nucleated and guided towards the basal end in a directed fashion to achieve gliding motility is unknown.

The invasive stages of most apicomplexans, including *T. gondii* and *C. parvum*, are highly polarized and contain an ensemble of apically localized structures known as the apical complex that participates in gliding motility[18–21]. Traditional electron microscopy (EM)[22,23] and cryo-electron tomography (cryo-ET)[19,20,24–26] studies have revealed the apical complex to contain a unique structure called the conoid, consisting of a cone-like arrangement of tubulin fibers and two associated rings above called the preconoidal rings (PCRs). Interestingly, when the parasite moves, the conoid extrudes through the apical polar ring (APR)[22,27–29] at the collar of the inner membrane complex (IMC; a network of flattened alveolar sacs underneath the parasite plasma membrane). Moreover, *T. gondii* Formin1 (FRM1), the sole and essential nucleator of F-actin for gliding motility[16], displays a ring-like localization above the conoid when observed by ultrastructure expansion microscopy (U-ExM)[30]. Conditional depletion of FRM1 simultaneously abolishes F-actin nucleation, conoid extrusion and parasite motility[16,30]. Depletion of MyoA, the myosin motor powering gliding motility and which itself is associated with the IMC, does not affect conoid extrusion but results in the accumulation of F-actin at the apical tip and defective motility[30]. Conversely, depletion of MyoH, the conoid-associated myosin motor, causes a defect in motility[31] and conoid extrusion[30], and results in F-actin leaking into the parasite body[30]. Thus, conoid extrusion seems to be intricately linked to F-actin flux for gliding motility. However, the underlying structure and mechanism have remained obscure, likely due to insufficient sample preservation and resolution inherent to conventional EM.

In this study, we use cryo-ET to investigate F-actin organization within motile, infectious *C. parvum* sporozoites that were experimentally released from spore-like oocysts[6]. *C. parvum* sporozoites are slender cells and have proven to yield excellent resolution for detailed in situ structural inspection[19,32]. Our tomograms reveal abundant F-actin at the apical tip of these parasites, enabling ultrastructural investigation of its apical nucleation site, subcellular localization during apico-basal translocation, and accumulation at the basal end. Using *T. gondii* tachyzoite mutants conditionally depleted for FRM1, we locate FRM1, and therefore the F-actin nucleation site, at the upper preconoidal ring. Together, we capitalize on the unprecedented resolution of cryo-ET to rigorously visualize and elucidate the path of apicomplexan F-actin in motile parasites.

## Results

### Ultrastructural observation of F-actin at the apical end of *C. parvum* sporozoites using cryo-ET

To elucidate the mechanistic basis of apicomplexan gliding motility, we performed cryo-ET on the apical end of freshly excysted *C. parvum* sporozoites (Fig. 1a–c). We observed numerous filaments that were oriented along the longitudinal axis of the parasite (Fig. 1d). Many of these filaments initiated near the preconoidal rings (PCRs) and ran

along the surface of the conoid (Fig. 1d); those long enough either entered the pellicular space between the IMC and the plasma membrane (Fig. 1e) or the cytoplasm (Fig. 1f). These filaments exhibited a diameter of 7–10 nm and a median length of 125 nm (Fig. 1g), resembling short F-actin predicted in apicomplexan parasites in some previous studies[10,12,33]. To verify the identity of these filaments, we imaged sporozoites that were treated with jasplakinolide[34], a drug that binds to and stabilizes F-actin. We observed a substantial increase in the abundance of filaments and that frequently resulted in rupturing of the plasma membrane, strongly suggesting that the observed filaments were indeed F-actin (Fig. 1h). Together, our tomograms of *C. parvum* sporozoites successfully revealed the ultrastructural arrangement of individual F-actin at the cell apical region.

### The upper PCR nucleates F-actin in *C. parvum*

The most apically positioned F-actin in our *C. parvum* tomograms consistently emanates from the upper PCR (i.e., <40 nm from it). These filaments were, on average, shorter compared to those positioned further down the parasite body within the pellicular space (median lengths of 86 vs. 143 nm), suggesting that they were likely newly initiated and still in the process of elongation (Fig. 2a). We, therefore, hypothesized that the upper PCR participates in F-actin nucleation and accordingly performed detailed structural analyses to test this. We found the upper ring to consist of 41 repeating subunits (Fig. 2b, c; from five parasites whose PCR subunits could be reliably counted). Each upper ring subunit was connected to a lower ring subunit by a linker at a constant length of ~18 nm (Fig. 2b). Since it was challenging to reliably count the individual lower ring subunits, we inferred 41 subunits for this ring as well based on the connections with the upper PCR. We observed that each PCR subunit was associated with a maximum of one actin filament, indicating the presence of a single nucleation site per subunit. We frequently observed filaments emanating from neighboring subunits (Fig. 2d – top panels). However, only 10 nucleation sites per cell on average showed proximal F-actin (with a maximum of 19 occupied subunits observed; Fig. 2d – bottom panel), suggesting that while all subunits appear equally capable, they do not necessarily all nucleate filaments synchronously.

We next performed subtomogram averaging on the PCRs to improve their resolution and investigate the relative spatial organization of F-actin ends. Averaging of 1097 subunits comprising both the upper and lower PCRs from 27 sporozoites resulted in a resolution of ~3.5 nm (Fig. 2e–1st and 3rd panels in black bounding boxes and Supplementary Movie 1). Averaging subunits from the two rings separately improved the resolution of the upper PCR to ~3 nm, indicating some flexibility between the two rings conferred by the linker (Fig. 2e–2nd and 4th panels, f, Supplementary Fig. 1, and Supplementary Movie 1). A "membrane anchor" density was seen anchoring the upper PCR into the plasma membrane, while a "protruding density" was observed extending from the lower region of the upper PCR and oriented outwards, facing the plasma membrane. When the subtomogram average was placed back into individual tomograms, the protruding density was the most proximal region of the PCRs to the F-actin ends at a median distance of 16.2 nm (Fig. 2g, h–right panel and Supplementary Movie 2). Notably, filament ends were variably positioned around the protruding density, either directly in front or in between two neighboring PCR subunits (Fig. 2h), suggesting some flexibility in the relative positioning of the actual nucleation site. However, we cannot rule out the possibility that this variability could be partly attributed to the mechanical compression of the parasite, an artifact of cryo-ET sample preparation by plunge freezing (see Methods– Quantifications, statistics, and reproducibility). Together, these data suggest that the protruding density of the upper PCR helps organize the nucleation sites for F-actin.

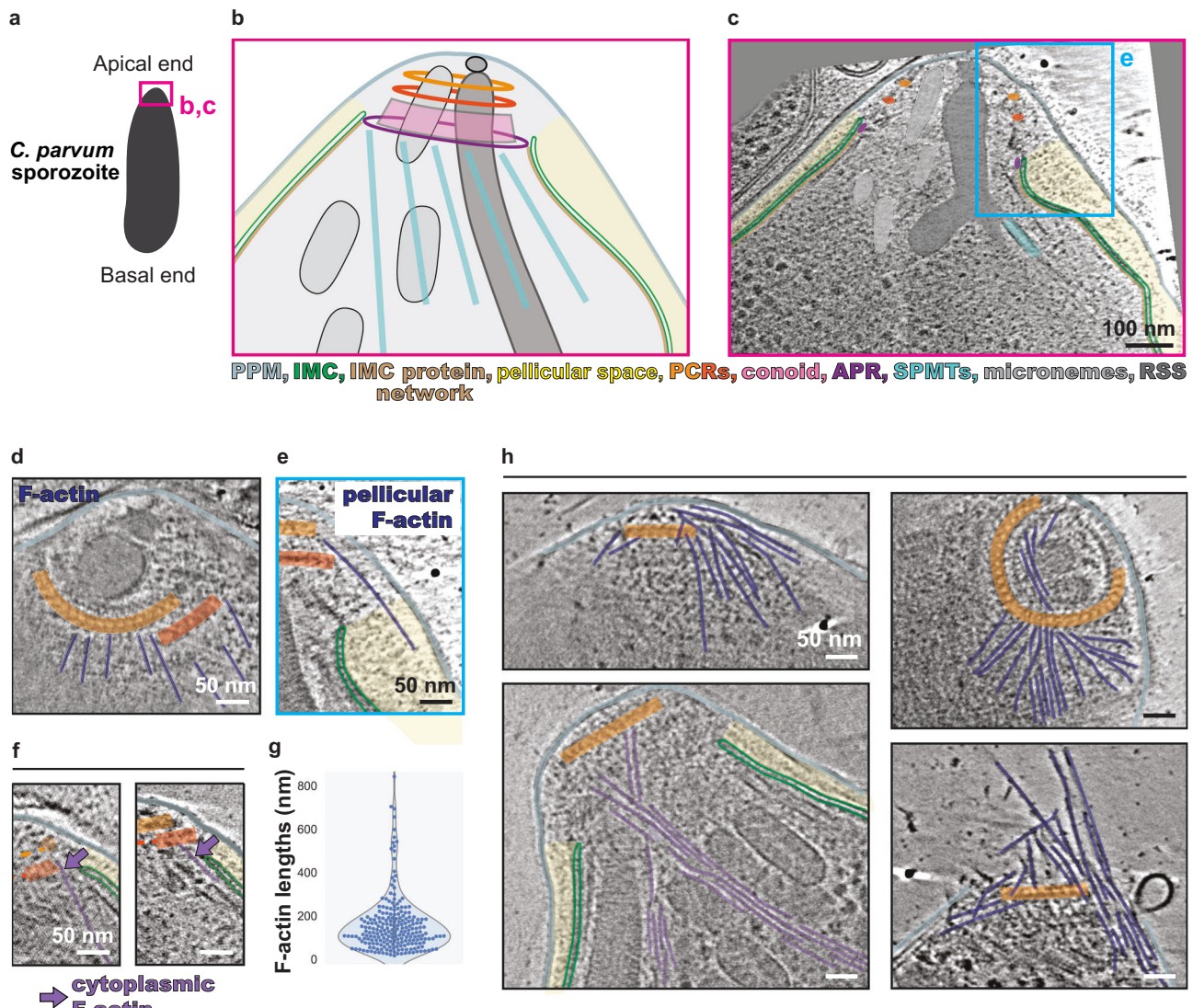

**Fig. 1 | Apical F-actin in *C. parvum* sporozoites. a** Schematic of a *C. parvum* sporozoite. The pink box denotes the region shown in panels **b** and **c**. **b** Schematic of the *C. parvum* apical end. **c** 2D slice of a tomogram of a *C. parvum* sporozoite, highlighting various structural features and organelles in the apical region. The cyan box denotes the region shown in panel **e**. **d**, **e** 2D slices from tomograms displaying F-actin around the PCRs and conoid, entering the pellicular space. **f** 2D slices from tomograms displaying F-actin around the conoid entering the cytoplasmic space. **g** Violin, box, and beeswarm plots of the lengths of all PCR-associated and pellicular F-actin in 20 *C. parvum* apical end tomograms. For the box plot underlying the beeswarm plot, the box limits denote the upper and lower quartiles while the whiskers denote the 1.5x interquartile range (this applies to all such plots in the main and supplementary figures). **h** 2D slices from tomograms of *C. parvum* treated with 1 µM (top panels and bottom left panel) or 5 µM (bottom right) of jasplakinolide. Unannotated images (without color overlays) for all relevant figure panels are included in Supplementary Fig. 10. PPM parasite plasma membrane, IMC inner membrane complex, PCRs preconoidal rings, APR apical polar ring, SPMTs subpellicular microtubules, RSS rhoptry secretion system. Scale bars are 100 nm in panel **c** and 50 nm in panels **d**–**h**.

## Conoid extrusion in *C. parvum* is linked to F-actin gating into the pellicular space

As previously mentioned, we observed apical F-actin entering either the pellicular space or the cytoplasm. The functional implications of these two kinds of localization and their determinants are unknown. Recently, disruption of conoid extrusion was linked to F-actin mislocalization within the *T. gondii* cell body using fluorescence microscopy[30], leading to the hypothesis that conoid extrusion might be related to subcellular positioning of F-actin for function. Using our tomograms, we investigated this potential link in *C. parvum* by analyzing precise F-actin localization in the context of conoid positioning. We found the much smaller conoid in *C. parvum* to be capable of extrusion, as reported for *T. gondii*. In 72 out of 82 parasites imaged (88%), the conoid was partially or fully extruded while seemingly under active F-actin generation. Such conoid displacements positioned the upper PCR from the APR at a distance ranging from ~150 nm in the fully extruded state to ~65 nm in the retracted state (Fig. 3a). Importantly, we noted a strong correlation between the localization of F-actin and the level of conoid extrusion. Specifically, parasites with a more extruded conoid displayed a higher proportion of F-actin in the pellicular space (Fig. 3b, c), suggesting that conoid extrusion and F-actin entry into the pellicular space are coupled. Interestingly, PCR-associated filaments that entered the cytoplasm were guided over the conoid and were often seen associating with the IMC collar via putative connectors (Fig. 3b–lower panel and Supplementary Fig. 2b, c). Consistent with the presence of such tethers, we observed the closest distance between cytoplasmic F-actin and the IMC collar to be fairly conserved (with a median of ~12 nm) (Fig. 3d). These cytoplasmic actin filaments also displayed, on average, much longer and variable lengths compared to those that entered the pellicular space (Fig. 3e). These architectural differences suggest that the cytoplasmic

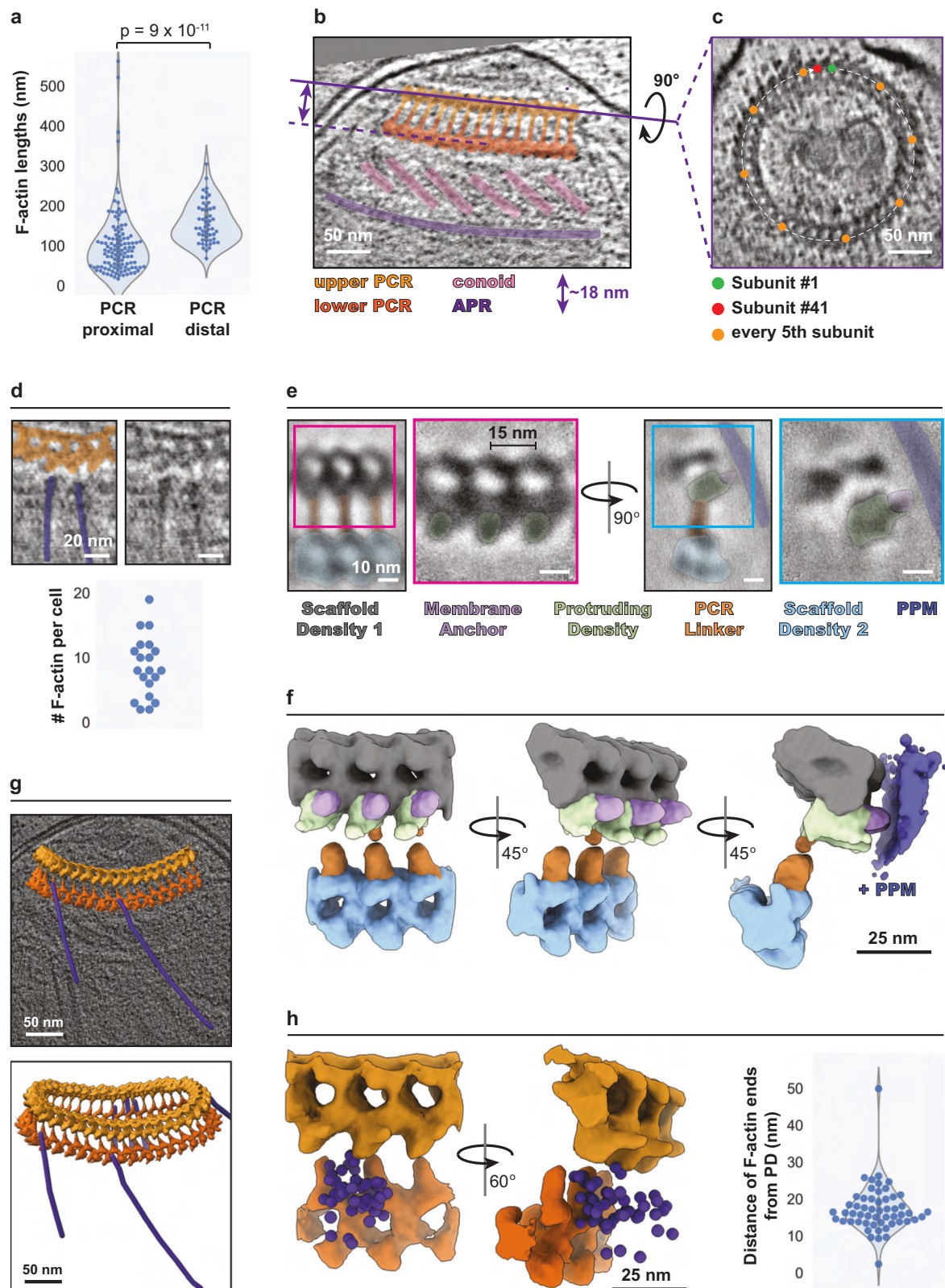

population of F-actin serves different functions from that of the pellicular population.

**An IMC surface filament in *C. parvum* suggests a structural basis for helical motility**

We observed that some pellicular F-actin had detached from the PCR and their lengths were consistently longer than those still associated with the PCR. This suggests that apically generated F-actin is detached from the PCR into the pellicular space to power gliding motility only after elongation to the desired length (Fig. 2a). *C. parvum* exhibits helical motility, indicating those detached F-actin may be translocated along a helical path down the cell length. We turned our attention to subpellicular microtubules (SPMTs) as they are known to have such helical arrangement and, therefore, could impart such F-actin

**Fig. 2 | F-actin is nucleated at the preconoidal rings. a** Violin, box, and beeswarm plots of the lengths of PCR-proximal vs PCR-distal F-actin that was not entering the cytoplasm. Filaments with ends <40 nm from the upper PCR were considered PCR-proximal and they either entered the pellicular space or were too short to determine their pellicular/cytoplasmic localization. PCR-distal filaments that entered the cytoplasm were extremely rare (n = 20 tomograms). A two-way Mann−Whitney U-test was performed to determine statistical significance. **b** Side view and **c** top-down view 2D slices of an apical end *C. parvum* tomogram, highlighting the PCRs and the presence of 41 subunits. The purple line in panel **b** denotes the cross-section observed in panel **c**. **d** 2D tomogram slices (with and without color overlays) showing F-actin ends near the upper PCR (top), and a beeswarm plot (bottom) showing the number of PCR-associated F-actin observed per cell (n = 20 cells). **e** Central slice of the combined *C. parvum* PCR subtomogram average (black boxes) and refined upper PCR subtomogram average (pink and cyan boxes) showing the front view (left) and side view (right) with color overlays. Note the gray Scaffold Density 1 (in **f**) has no color overlay here. **f** 3D segmentation of the refined subtomogram averages of *C. parvum* PCRs. The colors are the same as in panel **e**. Refer to Supplementary Movie 1. **g** The combined PCR subtomogram average fit back into a tomogram with segmented PCR-associated F-actin with (top) and without (bottom) the 2D tomogram slice shown. Refer to Supplementary Movie 2. **h** First and second panels show the locations of F-actin ends (purple spheres) around the PCRs. The refined subtomogram averages of the two PCRs were used for this purpose. The third panel shows violin, box, and beeswarm plots for the distance of F-actin ends from the protruding density of the associated upper PCR subunit (n = 20 tomograms). Unannotated images (without color overlays) for panels **b**, **c**, and **e** are included in Supplementary Fig. 11. Scale bars are 50 nm in panels **b**, **c**, and **g**, 20 nm in panel **d**, 10 nm in panel **e**, and 25 nm in panels **f** and **h**.

translocation. However, *C. parvum* sporozoites have only 2 to 3 long SPMTs that extend beyond the apical region (measuring >400 nm), while the vast majority are only 100–200 nm long (Supplementary Fig. 3a), making them unlikely candidates for guiding F-actin along the cell body. Instead, we discovered filaments on the external surface of the IMC in the pellicular space (Supplementary Fig. 3b). These IMC surface filaments (IMCSFs) were arranged with an overall left-handed helical twist like the underlying SPMTs and showed regular spacing between them (Fig. 4a, b), suggesting a tight spatial organization between IMCSFs and microtubules across the IMC. The IMCSFs exhibited an average interfilament spacing of ~40 nm and remained parallel to each other along the length of the cell body (Fig. 4c−left panel). As a control, computational randomization of their positions and orientations shows that their parallel organization was non-random (Fig. 4c−right panel).

We next investigated the spatial relationship between IMCSFs and pellicular F-actin. These F-actin were found to consistently run between and in parallel to the IMCSFs at a relatively fixed distance of ~20 nm (Fig. 4d, e−left panel). As a control, computational randomization of the positions and orientations of IMCSFs and F-actin resulted in the loss of this relationship (Fig. 4e−right panel). In contrast, we did not observe such an organization of cytoplasmic F-actin. Despite their overall helical arrangement, IMCSFs, particularly those further into the cell body, often appeared disorganized and kinked (Supplementary Fig. 3c). This could be an artifact of sample preparation (see Methods) or could be reflective of a flexible or dynamic nature of the filament. Moreover, F-actin organization was also perturbed further down the parasite body, where they crossed over into the adjacent inter-IMCSF space (Supplementary Fig. 3c), suggestive of potential interruptions in helical gliding. Overall, our data suggest that IMCSFs guide F-actin along a helical path down the cell body of *C. parvum* to promote helical gliding motility.

Given their association with F-actin, we sought to further characterize the structure of the IMCSFs pertinent to their organization and dynamics. IMCSFs were composed of repeating units and displayed two distinct conformations. The predominant conformation resembled interconnected loops, while the other, found especially near their apical and basal extremes, had a serrated sawtooth-like appearance (Fig. 4f). In some instances, the same filament could be seen transitioning between the two conformations (Fig. 4f). Separate subtomogram averaging of the "interconnected-loops" and "sawtooth" conformations revealed them to be composed of donut-shaped and S-shaped subunits, respectively, with connecting densities in between (Fig. 4g and Supplementary Movies 3, 4). Both conformations displayed two additional repeating densities with regular spacing – one extended towards the plasma membrane and the other towards the IMC; the latter seemed to tightly anchor the filaments into the IMC since the filaments remained associated with the IMC even in cells that accidentally lost their plasma membrane during sample preparation (Supplementary Fig. 3b). Taken together, the IMCSFs likely form channels that guide pellicular F-actin along a helical path.

## F-actin accumulates in the pellicular space at the basal end of *C. parvum*

Tomograms of the basal end of *C. parvum* sporozoites showed the accumulation of filaments in the pellicular space. These filaments formed a basal cap and displayed tight bundling by arranging into flat, multilayer sheets (Fig. 5a–e and Supplementary Figs. 4–6). Filaments were 7–10 nm in diameter with a median length of 156 nm. We hypothesized these filaments to be F-actin as their structure resembled similarly bundled F-actin in previous cryo-ET studies of different systems (Supplementary Fig. 4b, c)[35]. To experimentally test this idea, we imaged the basal ends of jasplakinolide-treated parasites by cryo-ET. In treated parasites, the filament length increased (median length of 234 nm; Fig. 5f), while their number and organization within the pellicular space remained unchanged. Thus, these basal filaments are likely accumulated F-actin used during gliding. Interestingly, F-actin accumulation is only observed beyond the termini of IMCSFs, which are apparently helically organized along the entire length of the parasite (Fig. 5b and Supplementary Fig. 4a). Actin filaments at this boundary were parallel to the associated IMCSF termini (Fig. 5c−left and Supplementary Fig. 4a). Since basally accumulated F-actin are similar in length to the apical filaments, run parallel to IMCSFs, and are found within the pellicular space, the simplest interpretation is that they are apically generated filaments that were guided by the IMCSFs along the length of the parasite.

The basal IMC almost completely sealed off the basal F-actin cap from the cytoplasmic space, which contained no observable F-actin (Supplementary Fig. 7a). Currently, the fate of basal F-actin is unclear. However, we noted an abundance of pore-like structures at the basal end of the IMC, each of which spanned the two membranes of the IMC, thus connecting the pellicular space with the cytoplasm. These pores may enable transport and recycling of actin across the IMC (Fig. 5b, d, g and Supplementary Fig. 7b). The pores were strictly enriched at the basal F-actin cap (Fig. 5b, d), and subtomogram averaging resolved an inner ring with a ~20 nm diameter, sufficient for monomeric globular actin or F-actin to pass through (Fig. 5h and Supplementary Fig. 7c). The average structure additionally showed faint central densities both within the pore and just outside on the cytoplasmic side (pink arrows in Fig. 5h). In the original tomograms, the central densities were visible in only a subpopulation of the pores (Supplementary Fig. 7b), suggesting that they were not integral pore components but instead transient components such as cargo passing through. Moreover, putative connecting densities were seen between the pores and nearby F-actin (Fig. 5g) that was retained in the average structure as well (Fig. 5h). Together, these observations allude to a potential role for the basal IMC pores in recycling actin into the cytoplasm that would likely have to be coupled to filament depolymerization (Supplementary Fig. 8a, b).

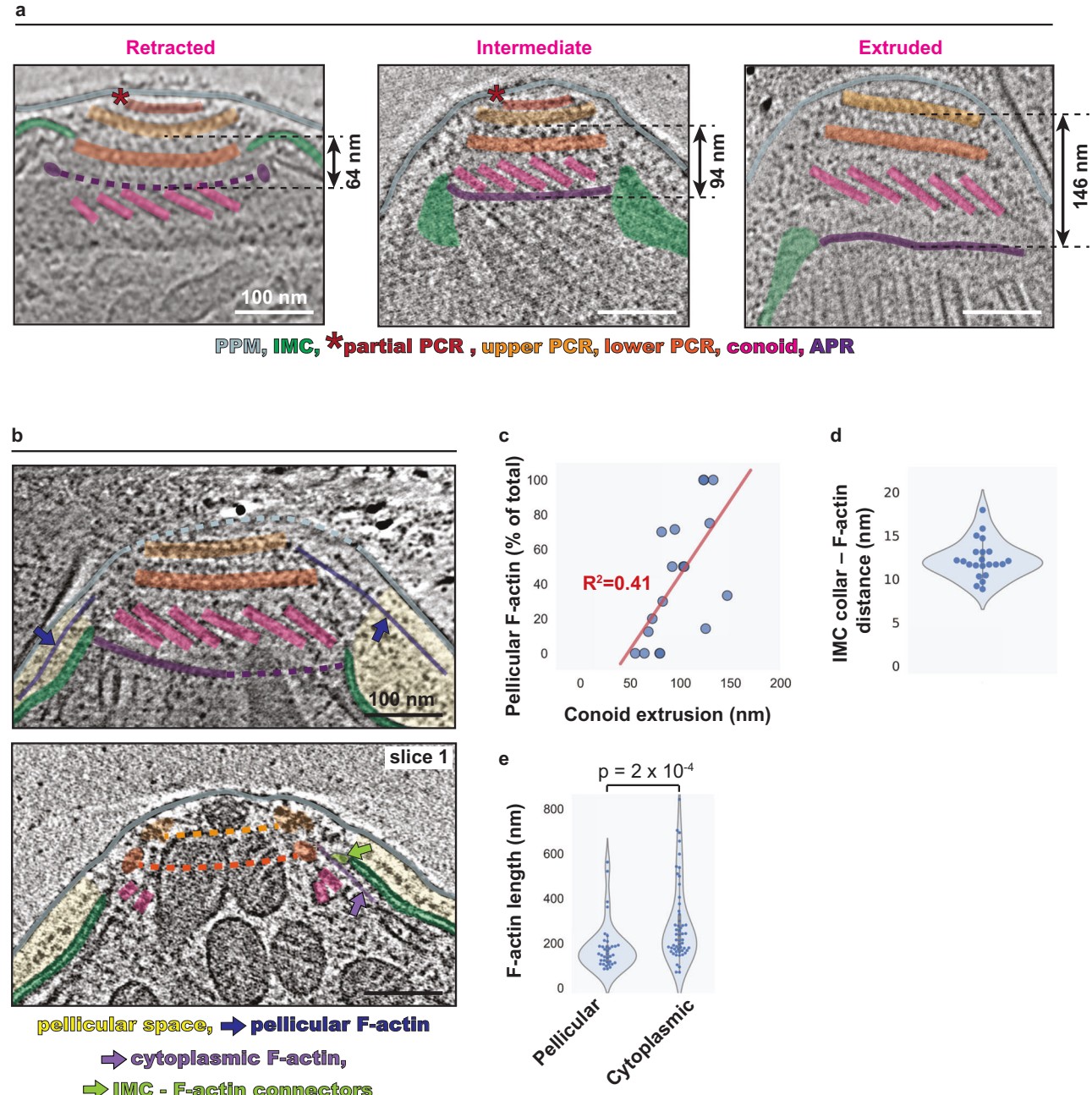

**Fig. 3 | Conoid extrusion permits pellicular F-actin gating in *C. parvum*. a** 2D slices of *C. parvum* tomograms displaying a retracted conoid (left), a partially extruded conoid (middle), and a fully extruded conoid (right). The measurement on each image represents the distance from the bottom of the upper PCR to the APR. **b** 2D slices from different tomograms displaying pellicular F-actin and cytoplasmic F-actin, both associated with the upper PCR. Note the difference in conoid extrusion between the two. In the bottom panel showing a parasite with a retracted conoid, a putative density is observed linking cytoplasmic F-actin to the IMC collar. An additional slice through the same tomogram is also shown in Supplementary Fig. 2a to unambiguously show the retracted conoid. **c** Scatterplot showing the extent of conoid extrusion (measured as the distance between the upper PCR and the APR) versus the proportion of PCR-associated F-actin that are pellicular. A regression line for the data is shown in red. **d** Violin, box, and beeswarm plots for the closest distance between cytoplasmic F-actin and the IMC collar ($n = 22$). **e** Violin, box, and beeswarm plots of cytoplasmic F-actin lengths and pellicular F-actin lengths (both PCR-associated). A two-way Mann–Whitney $U$-test was performed to determine statistical significance. Unannotated images (without color overlays) for panels **a** and **b** are included in Supplementary Fig. 11. Scale bars are 100 nm.

## The F-actin nucleator FRM1 localizes to the upper PCR in *T. gondii*

In addition to uncovering ultrastructural details of the apico-basal F-actin flux used for motility, our tomograms displayed F-actin emanating from the protruding density of the upper PCR, leading to the hypothesis that FRM1 localizes to this region. We tested this hypothesis using an established FRM1 knockdown system in *T. gondii*[16]. We first characterized the PCRs and F-actin nucleation in wildtype *T. gondii* tachyzoites, revealing 43 subunits for the PCRs (from three parasites whose PCR subunits could be reliably counted) and we again noted F-actin ends in their immediate proximity (Supplementary Fig. 9a, b). The *T. gondii* F-actin was similarly short, linear, and entered the pellicular space. Subtomogram averaging of the *T. gondii* PCRs (373 subunits from ten cells resolved to 5.3 nm resolution) revealed a similar

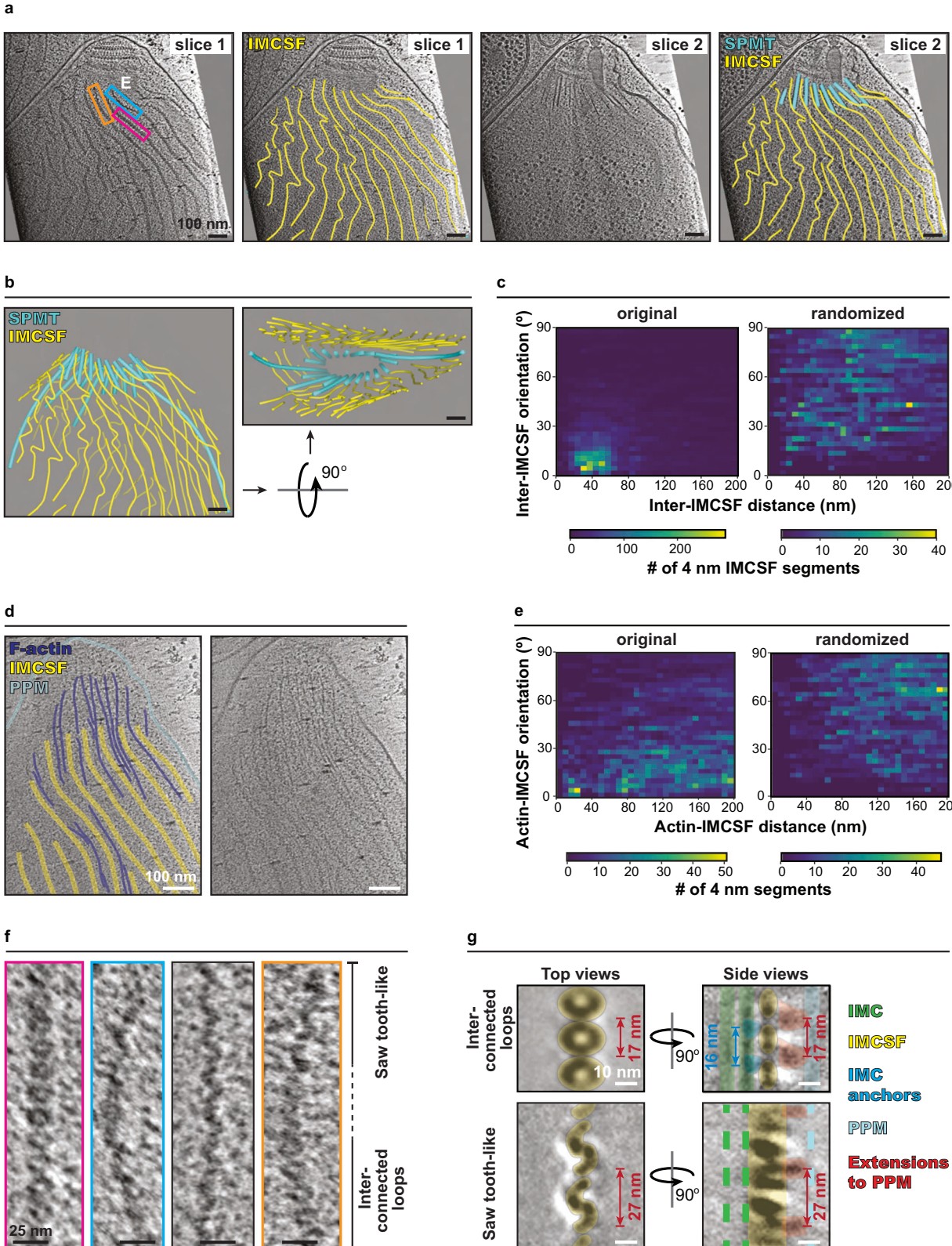

architecture to that of *C. parvum*, as evident from the overlay of the two structures (Fig. 6a, b and Supplementary Fig. 9c). The upper PCR subunit (separately refined to 4 nm resolution) displayed a similar protruding density. Intriguingly, the F-actin in *T. gondii* was more distanced from the protruding density, at a median distance of 27.2 nm compared to 16.2 nm in *C. parvum* (Fig. 6c, d). To address what might cause such difference in actin nucleation between the two species, we

conducted the bioinformatic analysis using *T. gondii* FRM1[16] (TgFRM1) to identify its homolog in *C. parvum* (CpFRM1; cgd6_4150). Both TgFRM1 and CpFRM1 are predicted to contain an N-terminal tetra-tricopeptide repeat (TPR)-like domain for protein–protein interaction and a C-terminal Formin Homology 2 (FH2) domain for actin nucleation. Interestingly, TgFRM1 contains a linker that is ~3200 amino acids longer than that of CpFRM1 and is predicted by AlphaFold2[36] to be

**Fig. 4 | IMC surface filaments in *C. parvum* provide helical tracks for F-actin.**
**a** 2D slices of a *C. parvum* tomogram showing the spatial arrangement of IMCSFs and SPMTs. The first two panels show the same tomogram slice with (right) and without (left) 3D filament segmentations. Similarly, the third and fourth panels show a different 2D slice with (right) and without (left) 3D filament segmentations. Colored boxes denote regions enlarged in panel **f**; **b** 3D segmentation of the SPMTs and IMCSFs from the *C. parvum* tomogram shown in panel **a**. **c** 2D histogram of inter-IMCSF distance versus orientation assessed using 4 nm IMCSF segments from the original tomogram (left) and a randomized control (right). **d** 2D slice of a tomogram displaying abundant apical F-actin arranged in between IMCSFs, with (left) and without (right) color overlay. **e** 2D histogram of F-actin-IMCSF distance versus orientation assessed using 4 nm filament segments from the original

tomogram (left) and a randomized control (right). **f** 2D slices of IMCSFs enlarged from the boxed regions in panel **a** (colored boxes) and from another cell (black box), displaying the loop conformation (pink box), sawtooth conformation (cyan and black boxes), and a section of filament transitioning between the loop and sawtooth conformations (orange box). **g** Subtomogram average of the IMCSF interconnected loop conformation (top) and sawtooth conformation (bottom). Note that the top and side views of the interconnected-loops conformation are obtained from two separate averages, while those of the sawtooth conformation are from the same average (see Methods). Unannotated images (without color overlays) for panel **g** are included in Supplementary Fig. 12. Scale bars are 100 nm in panels **a**, **b**, and **d**, 25 nm in panel **f**, and 10 nm in panel **g**.

more disordered (Fig. 6e). Such a significant difference in FRM1 size and architecture may contribute to the different F-actin nucleation distances observed between these two organisms.

We next performed subtomogram averaging of the PCRs in a conditional knockdown mutant for FRM1 in *T. gondii* (TgFRM1-iKD; 363 PCR subunits from ten cells) and compared the resulting average to that of wildtype (Fig. 6f, g and Supplementary Fig. 9c). In tomograms of TgFRM1-iKD, no F-actin was observed across all 28 cells imaged. While the overall structure of the two rings was similar between wild-type and TgFRM1-iKD (Fig. 6f), the difference map between the two averages suggests that TgFRM1 constitutes a portion of the protruding density (Fig. 6g). While the additional missing densities from our difference map could partially account for the other regions of FRM1 (Fig. 6g), it is clear that only a portion of the protein was resolved in the final subtomogram average of the wildtype PCRs, possibly due to its predicted flexible architecture (Fig. 6e). This interpretation is supported by the broad distribution of F-actin ends around the protruding density (Fig. 6d). Overall, these data are consistent with FRM1 localizing to the upper PCR in *T. gondii* and *C. parvum*, suggesting that this structure forms a flexible nucleation site for F-actin.

## Discussion

Nucleation of F-actin at the apical end of apicomplexan parasites and its apico-basal flux are essential for gliding and host cell invasion[14,16,17,37]. Biochemical and structural studies of *Toxoplasma* and *Plasmodium* actin suggest that their filaments are likely short and unstable in vivo, rendering them difficult to study[10,33]. These filaments were hypothesized to form and function within the pellicular space based on their interactions with the IMC-associated motor MyoA[38,39] and the translocation of surface adhesins on the parasite plasma membrane[40,41]. While this localization would be consistent with images obtained by fluorescence microscopy[16], definitive ultrastructural evidence has been lacking. Here, cryo-ET of *T. gondii* and *C. parvum* confirms the presence of such short, linear actin filaments in situ. Importantly, it allowed the analysis of subcellular localization and arrangement of F-actin with a resolution of individual filaments. We show pellicular F-actin in *C. parvum* and *T. gondii* and demonstrate the presence of such pellicular filaments at various positions along the length of the parasite body, supporting their role in directional parasite motility. Interestingly, we observe a greater abundance of apical F-actin in *C. parvum* sporozoites when compared to *T. gondii* tachyzoites. This difference may reflect their different life cycle stages, isolation methods, or species-specific dynamics (e.g., *C. parvum* sporozoites glide at higher speeds[6]).

The nucleation of F-actin close to the upper PCR suggests the localization of FRM1 to this ring, consistent with previous U-ExM experiments showing FRM1 labeling in the form of a ring in front of the conoid in *T. gondii*[30]. We tested this hypothesis by imaging and comparing wildtype and FRM1-depleted *T. gondii* tachyzoites and offer evidence that FRM1 localizes to the protruding density of the upper PCR. Due to the high structural similarities between the *T. gondii* and *C. parvum* PCRs, it is likely that FRM1 exhibits a similar localization in *C.*

*parvum* as well. PCRs have been found in all apicomplexans studied thus far and could therefore serve as the common hub for apical F-actin nucleation in this phylum. *T. gondii* and *C. parvum* FRM1 homologs are different in size and intriguingly nucleate actin filaments at different distances from the PCR. Our findings support a model in which the FRM1 N-terminal region (including the TPR domain) anchors the protein at the protruding density of the PCR while the C-terminal FH2 domain flexibly reaches out to nucleate F-actin (Fig. 6h). Consistent with this idea, the predicted structure of the TPR domain seems to fit well into one of the missing densities in the subtomogram average of TgFRM1-iKD PCR that constitutes a portion of the protruding density (Supplementary Fig. 9d). Since FH2 domains are known to typically function as dimers[42] and since F-actin was found on neighboring PCR subunits in our tomograms, it is likely that each PCR subunit harbors two copies of FRM1. However, further studies are needed to determine the exact stoichiometry and how FRM1 dimers may be constituted.

A recent study in *T. gondii* has proposed a model for how conoid extrusion might be gating F-actin into the pellicular space following its nucleation[30]. According to this model, conoid extrusion depends on force generation via the action of a conoid-associated myosin motor, MyoH, on the elongating F-actin. Once the conoid is extruded, F-actin can be properly translocated through the pellicular space to power gliding motility. Integrating published findings and our in situ structural observations, we propose the following mechanistic model, albeit speculative, for F-actin gating into the pellicular space (Supplementary Fig. 8c): Briefly, F-actin is nucleated at the PCRs, elongated, and channeled into the cytoplasm. The cytoplasmic F-actin is tentatively anchored at the IMC collar through putative connectors, while FRM1 continues to elongate the F-actin at the PCRs in a regulated fashion. Concomitantly, MyoH motors tethered to the conoid walk on the F-actin to power conoid extrusion, a process that may be assisted by other factors such as the Glideosome-Associated Connector (GAC) protein also known to associate with F-actin[41]. When the conoid extrudes, the PCRs are placed sufficiently above the IMC collar allowing the channeling of newly generated F-actin into the pellicular space to power motility (see Supplementary Fig. 8c for more details). It is interesting to consider how apicomplexans without a conoid, such as the merozoite stage of *Plasmodium*[20,43,44], properly localize F-actin into the pellicular space. The subtomogram average of the *P. falciparum* merozoite PCRs[44] reveals a different arrangement of subunits from that of *T. gondii*—the larger circumference of the lower PCR compared to the upper PCR causes it to protrude outward and likely provides the proper orientation for F-actin gating into the pellicular space even without the need for extrusion.

Upon entry into the pellicular space and elongation to the desired length, F-actin is translocated towards the cell basal end by MyoA motors to support forward parasite motion. However, the filaments must first detach from the upper PCR, a process that is potentially driven by MyoA itself or by GAC; the latter has been shown to also translocate in an apico-basal fashion in motile *T. gondii* cells[41] and is conserved in *C. parvum*. Our *C. parvum* tomograms reveal the

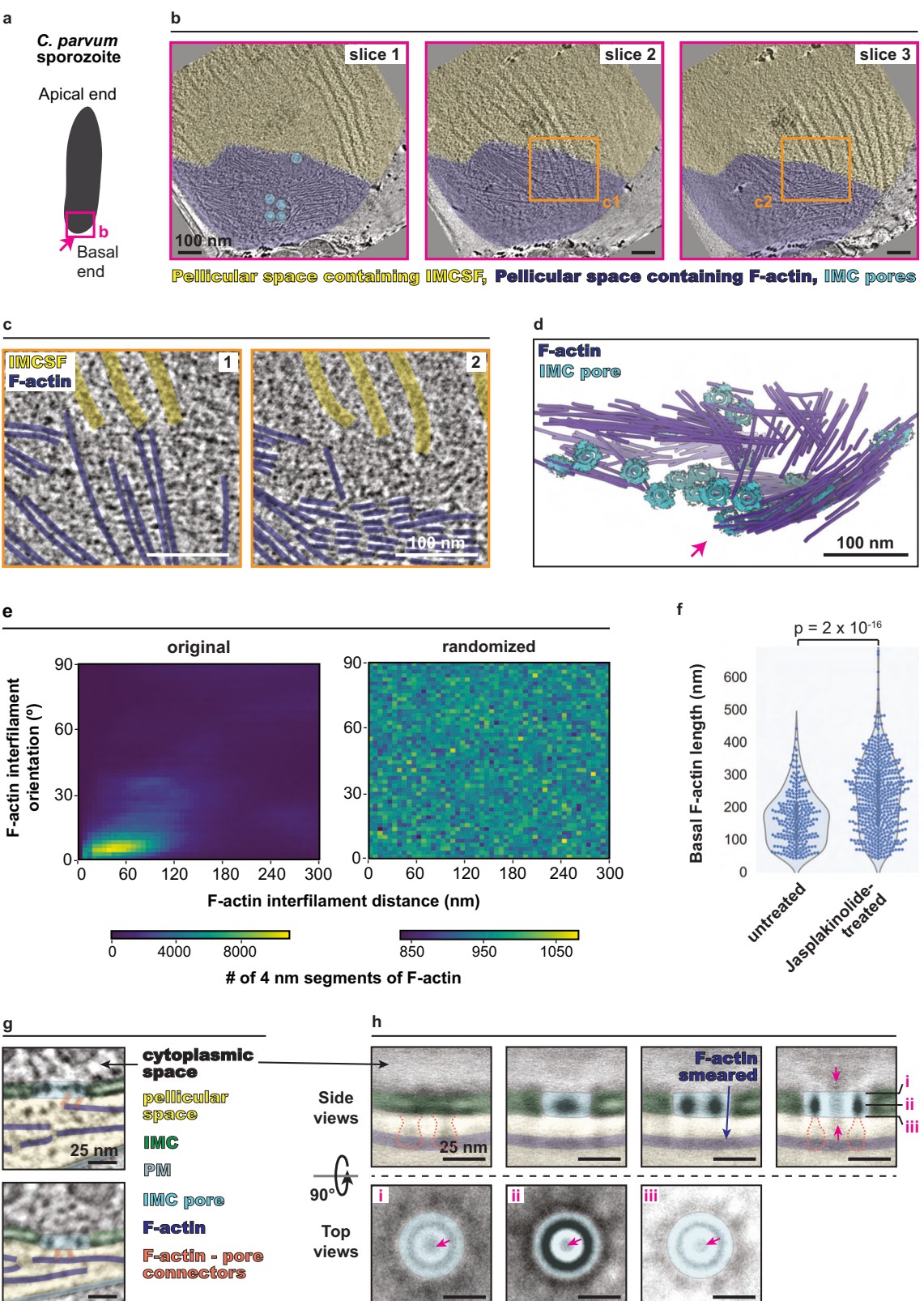

**a** *C. parvum* sporozoite — Apical end — Basal end — b

**b** slice 1, slice 2, slice 3 — 100 nm — c1, c2

Pellicular space containing IMCSF, Pellicular space containing F-actin, IMC pores

**c** IMCSF, F-actin — 1, 2 — 100 nm

**d** F-actin, IMC pore — 100 nm

**e** original / randomized — F-actin interfilament orientation (°) — F-actin interfilament distance (nm) — 0 4000 8000 / 850 950 1050 — # of 4 nm segments of F-actin

**f** Basal F-actin length (nm) — p = 2 x 10⁻¹⁶ — untreated, Jasplakinolide-treated

**g** cytoplasmic space, pellicular space, IMC, PM, IMC pore, F-actin, F-actin - pore connectors — 25 nm

**h** Side views — F-actin smeared — i, ii, iii — 25 nm — Top views — 90° — i, ii, iii

presence of IMCSFs on the pellicular face of IMC that could also potentially promote F-actin detachment—upon conoid extrusion, the upper PCR is positioned at ~200 nm from the IMCSF ends, which is the average length of the detached pellicular F-actin. In contrast, cytoplasmic F-actin was much longer on average. It is, therefore, possible that the IMCSFs provide a signal within the pellicular space that terminates the elongation of F-actin and instead promotes its detachment

to facilitate its role in motility. Apicomplexans are known to predominantly exhibit helical motion[5,6]. In the case of *Plasmodium* sporozoites *and Toxoplasma* tachyzoites, the SPMTs have been hypothesized to facilitate such a motion due to their helical arrangement along the length of the parasite body[17,45]. It is conceivable that the tight association of SPMTs with the IMC could help organize MyoA on the pellicular face of the IMC, thus providing helical tracks for F-actin

**Fig. 5 | F-actin accumulates at the basal end of *C. parvum*. a** Schematic of a *C. parvum* sporozoite. The pink box denotes the basal end, shown in panel **b. b** 2D slices of a *C. parvum* basal end tomogram displaying tight bundling of F-actin in the pellicular space, associated IMC pores, and IMCSFs. **c** Zoom-ins on the basal end tomogram shown in panel **b**, displaying bundling of F-actin into multilayered sheets and revealing the association between the basal termini of IMCSFs and basal F-actin. The two panels show different Z-slices (numbered 1 and 2) through the same region that is denoted by the orange boxes in panel **b. d** 3D segmentation of basal F-actin with the IMC pore subtomogram average placed back into the tomogram shown in panel **b**. The pink arrow indicates the basal tip of the parasite as in panel **a. e** 2D histogram of interfilament distance versus orientation of basal F-actin in untreated and jasplakinolide-treated parasites combined (left) along with a randomized control (right). $n = 2$ for untreated and $n = 2$ for jasplakinolide-treated parasites.

These parameters were assessed using 4 nm segments of the filaments. **f** Violin, box, and beeswarm plots of F-actin lengths in two untreated parasites and two jasplakinolide-treated parasites. A two-way Mann–Whitney *U*-test was performed to determine statistical significance. **g** 2D slices from a basal end tomogram displaying side views of an IMC pore tentatively associating with F-actin in the pellicular space. **h** Side views (top row) and top views (bottom row) of slices through the sub-tomogram average of the basal IMC pore generated by applying C8 symmetry (non-symmetrized average in Supplementary Fig. 7c). Pink arrows denote a faint density associated with the pore that is likely representing cargo proteins and the densities annotated with dashed orange outlines likely represent putative connectors between the pore and F-actin. Unannotated images for panels **b** and **c** are shown in Supplementary Fig. 4a and those of panels **g** and **h** are included in Supplementary Fig. 12. Scale bars are 100 nm in panels **b**–**d**, and 25 nm in panels **g** and **h**.

translocation down the parasite body. However, long SPMTs are not found in all apicomplexans or life cycle stages, as a majority of *C. parvum* sporozoite SPMTs do not extend beyond the apical region. Different mechanisms may be deployed to impart helicity to F-actin translocation in different organisms—in *C. parvum*, IMCSFs are helically arranged near both ends of the parasite and presumably along the entire cell body. Nonetheless, we note that we are technically limited from imaging the same IMCSF throughout the entire length of a parasite cell, and future studies using focused ion beam milling[32,46] (to thin the frozen parasites) in combination with montage cryo-ET[47] may help reveal additional details. Previously, conventional EM has reported "longitudinal ridges" in *C. parvum* sporozoites that were ~10 nm in diameter and resembling IMCSFs in their organization—they were arranged in parallel as they spiraled down the cell body in the pellicular space[48]. Initially misidentified as microtubules[49], they were later shown to not be tubulin-based using immunofluorescence[50]. These observations are consistent with IMCSFs constituting the ridges all along the parasite body and provide evidence that the kinking and partial disorganization of IMCSFs we see in the *C. parvum* tomograms are possibly artifacts of sample preparation. Longitudinal ridges are also present in gregarines, and 12 nm IMC filaments on the cytoplasmic face have been observed to influence the directionality of gliding[51]. Intriguingly, *Cryptosporidium* displays a closer phylogenetic affinity to gregarines than to coccidians, such as *Toxoplasma*[52]. F-actin regularly bisects the space between neighboring IMCSFs, suggesting that MyoA could be organized as helical longitudinal rows, regularly spaced between the IMCSFs. The IMCSFs might help to organize these F-actin tracks or simply act as "guardrails" for F-actin to achieve persistent helical gliding. The two identified IMCSF conformations (interconnected loops and sawtooth) share conserved structural features and membrane anchors, suggesting similarities in protein composition and possible interconversions between the two (two hypothetical models are illustrated in Supplementary Fig. 3d). It is unknown how each conformation may function to assist F-actin translocation but discovering the protein components of IMCSF in the future will facilitate this investigation. Interestingly, IMCSFs seem to be a species-specific adaption in *C. parvum*, as *T. gondii* tachyzoites do not show any similar features. It is, therefore, possible that the long SPMTs in certain species, including *T. gondii* are sufficient to guide F-actin in helical paths down the parasite body.

The apico-basal translocation of F-actin in *T. gondii* has been previously demonstrated to cause an accumulation of these filaments at the basal end using fluorescence microscopy[16]. Our tomograms of *C. parvum* reveal sequestration and extensive bundling of F-actin within the pellicular space as a "basal cap". F-actin-binding factors may contribute to the accumulation and bundling of these filaments. For instance, F-actin stabilizing proteins, GAC, and coronin, have been shown to accumulate at the basal end of invading *T. gondii* tachyzoites[41,53]. We observed pores embedded within the IMC that are specifically positioned at the basal F-actin cap, occasionally showing tentative associations with F-actin. They display variable cargo-like central densities, suggesting a possible role in recycling the accumulated F-actin back into the cytoplasm, possibly in the form of monomeric globular actin. Such actin recycling appears to be tightly regulated since the tomograms reveal a stark difference in F-actin abundance across the IMC. The absence of cytoplasmic F-actin at the basal end also suggests that monomeric globular actin could be sequestered by actin-binding proteins such as profilin[54,55] immediately following F-actin depolymerization either before or after transport across the IMC. F-actin dynamics and recycling at the basal end of *T. gondii* could potentially be different from that of *C. parvum* due to a possible large opening in the basal IMC[56,57].

In summary, this study capitalizes on native, in situ structural observations to build a comprehensive model for the life cycle of F-actin, enabling the gliding motility of apicomplexan parasites. The details and predictions of this model can now be systematically tested in various apicomplexans using mutants, an effort that will help understand how this unique and important mechanism for motility has evolved across the phylum.

## Methods

### Preparation of *C. parvum* sporozoites

*C. parvum* oocysts, purchased from Bunch Grass Farm (Deary, Idaho, USA), were prepared as previously described[19]. Oocysts were washed for 10 min in bleach at 4 °C and excystation was triggered by 0.8% sodium taurodeoxycholate (Sigma-Aldrich, St. Louis, MO, USA, Cat# T0875) for 10 min at 16 °C and subsequent incubation at 37 °C for 1 h. Parasites were then resuspended in PBS or DMEM-10 (consisting of DMEM—ThermoFisher, Waltham, MA, USA, Cat# 10313039; 10% FBS—Atlanta Biologicals, Flower Branch, Georgia, USA, Cat# S11550; L-Glutamine—Gemini Bio Products, CA, USA, Cat# 400-106; Penicillin–Streptomycin—ThermoFisher, Cat# SV30010). 10 nm colloidal gold fiducials (Ted Pella, Redding, USA) were added to the suspension (for alignment purposes during tomogram reconstruction from tilt series). Approximately, 4 µl of suspended cells (estimated at ~$4 \times 10^6$ sporozoites) were applied onto Quantifoil 200 mesh copper R2/2 holey carbon EM grids, excess liquid blotted away, and plunge frozen in a liquid ethane/propane mixture (pre-cooled with liquid nitrogen) using an EM GP2 automatic plunger (Leica Microsystems, Wetzlar, Germany)[58]. The blotting chamber was set to 95–100% relative humidity at 37 °C and blotting was done from the sample side of the grid using Whatman filter paper #1. Plunge-frozen grids were subsequently loaded into autogrid c-clip rings (Thermo Fisher). The autogrid containing frozen grids were stored in liquid nitrogen and maintained at ≤−170 °C throughout storage, transfer, and cryo-ET imaging.

### Drug treatments for *C. parvum*

After excystation (prior to plunge freezing), sporozoites resuspended in DMEM-10 were incubated at 37 °C with 1 µM jasplakinolide (Sigma-Aldrich, Cat# J4580) for 40 min (or with 5 µM jasplakinolide for 65 min).

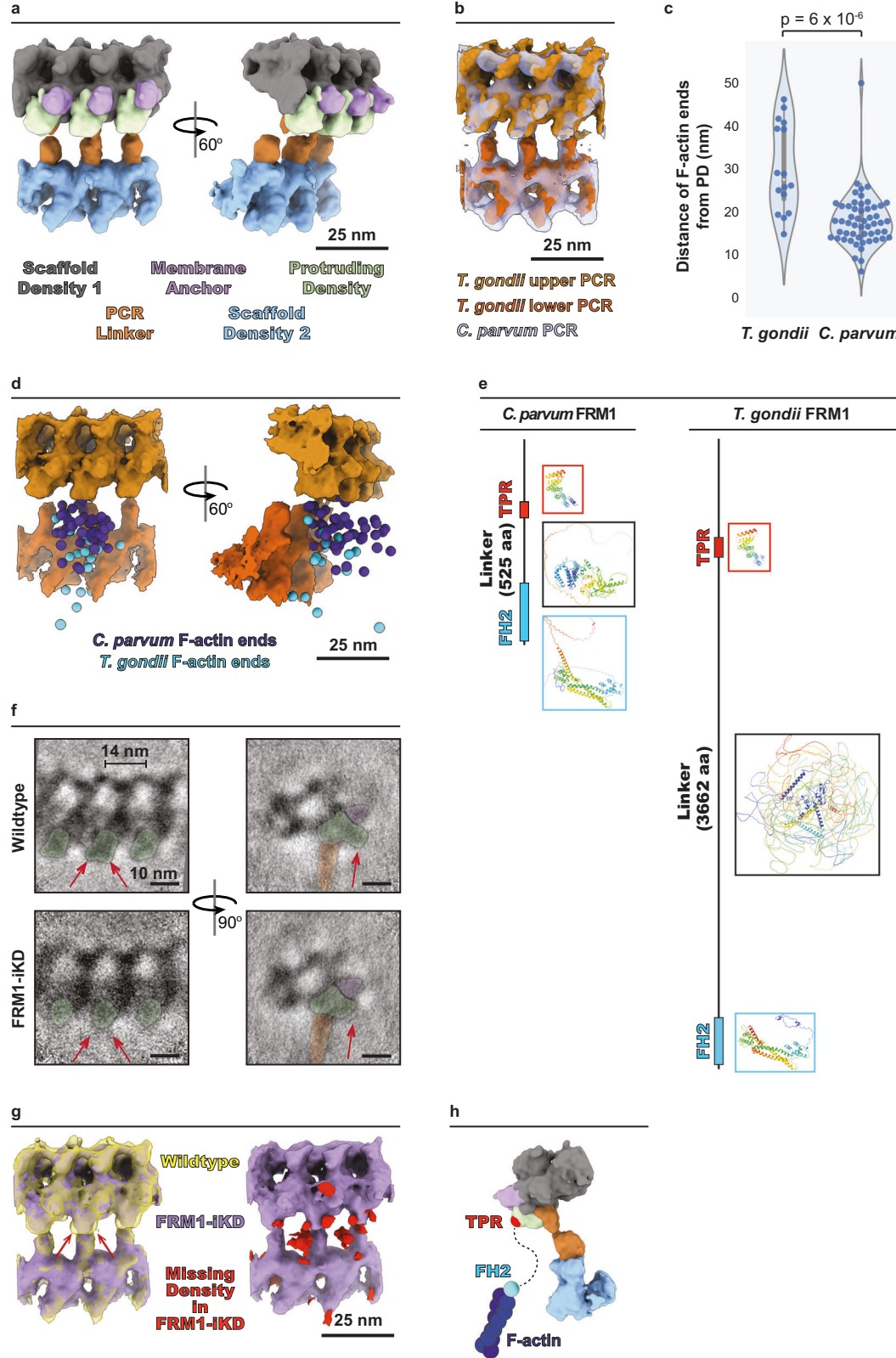

## Preparation of *T. gondii* tachyzoites

*T. gondii* tachyzoites from the RH strain were cultivated and prepared for cryo-ET as previously described[19]. Tachyzoites were cultivated in a monolayer of human foreskin fibroblasts (HFF – ATCC, CRL 1634) with DMEM-10 media supplemented with 5% fetal calf serum (FCS) and 2 mM glutamine. Upon egress, extracellular parasites were collected and resuspended in PBS or DMEM-10 with 10 nm gold fiducials before loading ~$4 \times 10^6$ tachyzoite (in a 4 μl suspension) onto a grid. For the FRM1-iKD strain, parasites were pretreated with 500 μM of Auxin (IAA) for 48 h prior to mechanical isolation of tachyzoites to allow for full depletion of FRM1 via the C-terminal mAID fusion[16]. Isolated tachyzoites were subsequently prepared identically to the wildtype parasites.

**Fig. 6 | FRM1 localizes to the upper PCR in *T. gondii*. a** 3D segmentation of the refined subtomogram averages of *T. gondii* PCRs. Note the similar densities to the PCRs of *C. parvum*. **b** Overlay of the refined subtomogram averages of the PCRs from *C. parvum* and *T. gondii*. *C. parvum* PCRs are made transparent to show the large similarities between the two organisms. **c** Violin, box, and beeswarm plots of the distances from the protruding density of the upper PCR to the F-actin ends in *T. gondii* and *C. parvum*. A two-way Mann−Whitney *U*-test was performed to determine statistical significance. **d** Two views depicting the locations of F-actin ends in relation to the PCRs in *T. gondii* and *C. parvum* to highlight the difference in the spatial arrangement of F-actin ends in the two organisms. F-actin ends are represented by purple spheres for *C. parvum* and by cyan spheres for *T. gondii*. **e** Domain architecture of FRM1 from *C. parvum* and *T. gondii*, along with the AlphaFold2 prediction for the structure of each domain. Everything is shown to scale. Note the differences in size and order between the FRM1 linkers from *C. parvum* and *T. gondii*. **f** Front and side view 2D slices from the subtomogram averages of *T. gondii* upper PCR from wildtype (top) and FRM1-iKD (bottom) strains. Colors for overlay are the same as in panel **a**. Note the missing densities at the protruding density in the FRM1-iKD, denoted by red arrows. **g** Overlay of subtomogram averages of the PCRs from wildtype and FRM1-iKD *T. gondii* (left) and the 3D difference map between the two (right), showing missing densities from the mutant subtomogram average in red. **h** A model of TgFRM1 architecture at the PCR – the TPR domain (red sphere) anchors to the protruding density while the flexibly linked FH2 domain (cyan sphere) hangs in front of the PCRs and nucleates F-actin. Unannotated images (without color overlays) for panel **f** are included in Supplementary Fig. 12. Scale bars are 25 nm in panels **a**, **b**, **d**, and **g**, and 10 nm in panel **f**.

## Cryo-electron tomography (cryo-ET*)*

Cryo-ET was performed on a Thermo Fisher Krios G3i 300 keV field emission cryo-transmission electron microscope. Dose-fractionated imaging was performed using the SerialEM software[59] on a K3 direct electron detector[60] (Gatan) operated in electron-counted mode. Motion correction of images was done using the Alignframe function in IMOD[61]. Imaging was done using a Volta phase plate[62] to increase contrast without high defocus, and the Gatan Imaging Filter (Gatan) with a slit width of 20 eV to increase contrast by removing inelastically scattered electrons[63]. After initially assessing cells at lower magnifications for suitability of ice thickness and plasma membrane integrity, tilt series were collected with a span of 120° (−60° to +60°; bi-directional scheme) with 2° increments at a magnification of 33,000x (with a corresponding pixel size of 2.65 Å) and a defocus range of −1 to −4 μm. Each tilt series was collected with a cumulative dose of around 140 e$^-$ Å$^{-2}$. Once acquired, tilt series were aligned using the 10 nm colloidal gold as fiducials and reconstructed into tomograms by our in-house automated computation pipeline utilizing the IMOD software package[61]. See Supplementary Table 1 for cryo-ET data collection and processing parameters.

## Quantifications, statistics, and reproducibility

We obtained a total of 228 *C. parvum* tomograms (119 untreated from three biological replicates, 109 jasplakinolide-treated from two biological replicates) and 128 *T. gondii* tomograms (100 wildtypes from three biological replicates, 28 FRM1-iKD from one biological replicate). For each sample, multiple frozen grids were imaged over several sessions, each spanning multiple days. Each of the following quantifications is from a subset of these tomograms, chosen based on best contrast but otherwise random. We note that ~50% of *T. gondii* parasites imaged had a broken plasma membrane, resulting in a greater resolution of intracellular features compared with completely intact tachyzoites. We also note that parasites flattened on the grid. However, this flattening, which was predominantly in the blotting direction and therefore probably caused by the blotting force, was mostly homogeneous in this direction and did not adversely affect the shape and organization of various structures and organelles, except for some flattening of the PCRs, APR, and the conoid. Flattening could have added subtle variations to the relative positions of features, but their organizational patterns were evident despite the presence of such potential effects. Importantly, we do not expect it to adversely affect the positioning of F-actin tips with respect to the PCRs, causing only subtle variations, if any, along the blotting direction.

## Apical F-actin length

IMOD models were generated along the length of every visible actin filament in the apical region of 20 *C. parvum* tomograms. Filaments with the following architecture were distinguished as separate objects—(1) PCR-associated F-actin (filaments whose ends were within 40 nm of the upper PCR) that either entered the pellicular space or were too short to determine their pellicular/cytoplasmic localization; (2) PCR-associated F-actin that entered the cytoplasmic space; and (3) PCR-detached F-actin that entered the pellicular space. PCR-detached filaments that entered the cytoplasm were extremely rare. IMOD models were generated with filament contours, where each filament was manually and carefully traced in 3D through multiple slices/angles in the tomograms to help accurately estimate filament lengths. Our tomograms offered sufficient contrast to discern filament ends in 3D. Representative 2D slices cutting through the 3D tomograms (for example, Figs. 1d, h, 4d that show annotated filaments) were chosen for display purposes. It is noteworthy that while annotating single 2D slices from tomograms can result in false positives and false negatives, our use of entire tomograms to examine filament continuity into neighboring slices helped to significantly reduce such instances. To further ensure accurate identification of actin filaments, two people (M.M. and S.K.M.) independently examined the results.

## Distance from PCR protruding density to F-actin end

After the generation of the *C. parvum* PCR subtomogram average, each F-actin end (a model point generated in IMOD) was mapped onto the subtomogram average based on the alignment information from subtomogram averaging. The subtomogram average was visualized in UCSF ChimeraX[64] and a marker was placed on the protruding density. After the alignment of F-actin ends onto the subtomogram average, the distance from the protruding density marker and each F-actin end was calculated.

## Inter-IMC surface filament distance and orientation

IMOD models were generated along the length of every visible IMCSF from 10 *C. parvum* sporozoites. Each filament was divided into 4 nm segments and assigned an orientation based on the vector between that segment and the next segment along the filament. For interfilament distances, the distance between a given segment and all the other segments from all other filaments was calculated. For orientation analysis, the angle between the orientation vector of a given segment and the orientation vector of other segments from all other filaments was calculated. As a control, the positions and orientations of filaments were randomized prior to the calculation of the interfilament distances and orientations (again using 4 nm segments). The python scripts to run these analyses are available at https://github.com/GrotjahnLab/measure_models/tree/0.1[65] and are free to use or modify with a BSD license. The implementation is very similar to that of a previous publication[66].

## F-actin-IMC surface filament distance and orientation

IMOD models generated along apical F-actin and IMCSFs were divided into 4 nm segments and assigned an orientation as described above. For distances between F-actin and IMCSFs, the distance between a given IMCSF segment and all F-actin segments was calculated. For orientation analysis, the angle between the orientation vector of a given IMCSF segment and the orientation vector of all F-actin segments was calculated. As a control, all filament positions and orientations were

randomized prior to the calculation of the interfilament distances and orientations for every 4 nm segment.

**Distance between the upper PCR and the APR to estimate conoid extrusion.** In IMOD, four pairs of points were placed equidistantly around the conoid of 20 parasites. One point was placed at the bottom of the upper PCR subunit, and another point was placed on the APR below. The average of these four distances was then calculated.

**Proportion of pellicular F-actin.** The number of PCR-associated pellicular F-actin divided by the total PCR-associated F-actin per cell.

Quantifications were performed on IMOD-generated models. All model files were exported for analysis and plotting using Pandas, Numpy, Matplotlib, and Seaborn libraries in Python 3.7 and Python 3.8. The two-way Mann–Whitney *U*-test was performed to determine statistical significance between populations with non-normal distributions.

### Subtomogram averaging of the preconoidal rings
All subtomogram averaging parameters are shown in Supplementary Tables 2–4. All subtomogram alignment and averaging were performed using Dynamo[67,68]. In IMOD, *C. parvum* and *T. gondii* tomograms were inspected manually for preconoidal ring structures. Subtomogram boxes were centered in the middle of the linker connecting the upper and lower PCR subunits and manually oriented to the same orientation among PCR ultrastructures. For *C. parvum*, we selected and manually oriented 1097 PCR subunits from 27 tomograms. For *T. gondii*, we selected and manually oriented 373 PCR subunits from nine wild-type tomograms and 366 PCR subunits from 11 FRM1-iKD tomograms. Model points and manual orientations were imported into Dynamo using custom MatLab scripts. Subtomograms were cropped with a box size of (360 voxels)$^3$ (corresponding to (95.4 nm)$^3$) to account for the upper and lower PCR subunits. To generate an initial reference, subtomograms were averaged using manual orientations and without any computational alignment. An ellipsoid mask that covered both upper and lower PCR subunits was used. For subtomogram alignment, subtomograms were downsampled to bin 4 dimensions and an angular sampling of 5° for 30° total for all Euler angles and a translational search of 2.6 nm in all directions was performed. A lowpass filter of 4–6 nm was applied to avoid overfitting. The resulting average was then used as a starting reference for the next iteration, and this process was repeated for three iterations in total.

To independently refine the subtomogram average of the upper and lower PCR subunits from each sample, Dynamo subboxing was used to re-crop subtomograms centered on either the upper PCR or lower PCR subunit. For both structures, a box size of (200 voxels)$^3$ (corresponding to (53 nm)$^3$) was used. An initial reference was generated by averaging subtomograms with previously computed alignments from the entire PCR subunit. An ellipsoid mask that covered the entire upper PCR subunit or the entire lower PCR subunit was used. For subtomogram alignment, subtomograms were downsampled to bin 4 dimensions and an angular sampling of 2° for 12° total for all Euler angles and a translational search of 8 nm in all directions (to account for flexibility between the upper and lower PCR subunits) was performed. A lowpass filter of 4–6 nm was applied to avoid overfitting. The resulting average was then used as a starting reference for the next iteration, and this process was repeated for three iterations in total.

To generate Fourier shell correlation (FSC) plots, the subtomograms were divided into two half-sets, independently averaged using initial orientations, and aligned and averaged independently using identical alignment parameters. After subtomogram averaging of the two independent half-sets, the resulting averages were aligned and their FSC was computed. The resulting resolution is reported at the 0.143 FSC threshold.

### Subtomogram averaging of the IMC surface filaments
All subtomogram averaging parameters are shown in Supplementary Table 5. In IMOD, IMCSF centerlines were traced and saved as independent models. In Dynamo, a custom MatLab script was used to import model points, create a "Filament with Torsion" model, and distribute points every 14 nm along each filament centerline (corresponding to one point per IMCSF subunit). This yielded a total of 6709 subtomograms from 24 tomograms. Subtomograms were cropped with a box size of (160 voxels)$^3$ (corresponding to (42.4 nm)$^3$; ~3 IMCSF subunits). Subtomograms from each tomogram were aligned and averaged independently of those from other tomograms to determine IMCSF quality per tomogram. To generate initial models, subtomograms from each tomogram were averaged with initial filament orientations and without computational alignment. Subtomograms from each tomogram were then aligned with an angular sampling of 4° for 24° total in the cone range (Euler angles Z and X) and an azimuth sampling (Euler angle Z') of 10° for 60° total, as well as a translational search of 7.5 nm in all directions. A cylindrical mask that covered the width of the IMCSF and the central subunit plus half of each neighboring subunit was used for alignment. A lowpass filter of ~3 nm was applied during subtomogram alignment. The resulting average was then used as a starting reference for the next iteration, and this process was repeated for three total iterations.

Upon creating an IMCSF average for each tomogram, three conformations were resolved: the IMCSF top view, side view with visible IMC cross-section, and "sawtooth" conformation. The best average of each conformation was used as an initial reference for multireference alignment. Here, all 6709 subtomograms were aligned against three starting references (IMCSF top view, side view with the IMC cross-section, and "sawtooth" conformation) using identical parameters as described above and assigned to the conformation to which each subtomogram scored the highest cross-correlation score. In the end, 5421 subtomograms were assigned to the top view class; 359 subtomograms were assigned to the side view with the IMC; and 929 subtomograms were assigned to the "sawtooth" conformation. Thus, the top and side views of the interconnected-loops conformation shown in Fig. 4g are two separate averages generated from two different sets of subtomograms; the two sets were oriented somewhat orthogonally relative to each other under the electron beam while imaging. For visualization of the "sawtooth" conformation (Fig. 4f, g), subtomograms were re-cropped and averaged with a box size of (280 voxels)$^3$ (corresponding to (74.2 nm)$^3$). The top and side views of the sawtooth conformation, unlike the interconnected-loops conformation, were obtained from the same average whose constitutive subtomograms were predominantly oriented close to the top view under the electron beam while imaging.

### Basal F-actin analysis
Four *C. parvum* basal end tomograms (two untreated, two jasplakinolide-treated) were identified based on the visibility of the basal F-actin cup in the field of view. First, tomograms were imported into the EMAN2 tomogram annotation workflow[69] and a neural network was trained to recognize and segment density corresponding to F-actin. Segmented F-actin was subsequently imported into UCSF Chimera[70] where the segmentations were cleaned up using the volume eraser tool to erase false positive noise. The segmentations were then imported into Amira software (Thermo Fisher), where a filament tracer tool was used to identify and trace individual filaments of F-actin. During this step, cylindrical correlation was used with a cylinder length of 40, angular sampling of 5, mask cylinder radius of 5.5, and outer cylinder radius of 4. Trace correlation lines were generated with a minimum seed correlation of 160. F-actin traces were extracted from Amira and analyzed using a computational toolbox[71] for quantitative analysis of interfilament orientation and distance parameters. These values were plotted using Matplotlib and Seaborn libraries in Python 3.7.

## Subtomogram averaging of the basal IMC pore

All subtomogram averaging parameters are shown in Supplementary Table 6. 235 IMC pores from 10 untreated parasites and 108 IMC pores from eight jasplakinolide-treated parasites were selected and manually oriented to similar orientations in IMOD. Particle positions and orientations were imported into Dynamo using custom MatLab scripts. Subtomograms were cropped with a box size of (320 voxels)$^3$ (corresponding to (84.8 nm)$^3$). To generate an initial reference, subtomograms were averaged using manual orientations and without any computational alignment. An ellipsoid mask was used that covered the entire pore and part of the IMC. For subtomogram alignment, subtomograms were downsampled to bin 4 dimensions and an angular sampling of 2° for 12° total Z and X Euler angles, 11.25° angular sampling for 45° total for the Z' Euler angle, and translational search of 5.3 nm in all directions was performed. A lowpass filter of 4–6 nm was applied to avoid overfitting. The resulting average was then used as a starting reference for the next iteration, and this process was repeated for three iterations in total.

Upon observing the eightfold arrangement of densities throughout the subtomogram average, we applied C8 symmetry resulting in a total of 2744 subtomograms. The same initial reference and mask were used for subtomogram alignment with symmetry. For alignment, subtomograms were downsampled to bin 4 dimensions and an angular sampling of 2° for 12° total for all Euler angles, and a translational search of 2.6 nm in all directions, was performed. A lowpass filter of 3–4 nm was applied. The resulting average was then used as a starting reference for the next iteration, and this process was repeated for three iterations in total.

## Identification of the *C. parvum* formin1 gene

On the ToxoDB, we performed a BLASTP (Protein Blast) search using TgFRM1 (TGME49_206430) against the nonredundant (nr) protein database. The top hit for *Cryptosporidium* was cgd6_4150, a hypothetical FH2 domain-containing protein.

## Structure prediction with AlphaFold2

The domains of CpFRM1 and TgFRM1 (TPR, linker, and FH2 domains) were submitted individually to the ColabFold (AlphaFold2.ipynb) Google Notebook since jobs on proteins over ~1000 amino acids were generally too large for the server. The following amino acids for each protein were submitted: for CpFRM1, amino acids 1–640 (TPR domain-containing region), 561–1638 (linker and FH2 domain-containing region); for TgFRM1, amino acids 1–1040 (TPR domain-containing region) and 4161–5009 (FH2 domain-containing region). The TgFRM1 linker region, containing amino acids 940–4240, was submitted as an AlphaFold2 job on a local workstation to avoid timing out of the online server.

## Generation of a difference map between *T. gondii* WT and FRM1-iKD PCR structures

Subtomogram averages of the entire PCR and the refined upper and lower PCRs from WT and FRM1-iKD *T. gondii* were imported into UCSF ChimeraX. First, the entire PCR structures of WT and FRM1-iKD were aligned using the command "fit model 1 inMap model 2", where "model 1" and "model 2" were the two PCR structures. The refined upper and lower PCR structures were similarly aligned to their respective regions in the entire PCR subtomogram average. Once aligned, the command "volume subtract" was used to subtract the FRM1-iKD PCR density from the WT PCR density.

## Figure presentation, modeling, and segmentation

Tomograms were oriented in 3D using IMOD's Slicer window such that the desired tomogram section was in view for presentation. To enhance contrast, 5–10 layers of voxels spanning thicknesses of 5.3–10.6 nm were averaged around the section of interest. Figures were prepared using Adobe Illustrator 2022. Segmentations of subtomogram averages were done utilizing the Segger function in ChimeraX. Visualizations and animations of subtomogram averages were generated using ChimeraX.

## Reporting summary

Further information on research design is available in the Nature Portfolio Reporting Summary linked to this article.

## Data availability

The data that support this study are available from the corresponding authors upon request. Representative tomograms are available in the Electron Microscopy Data Bank (EMDB) under accession codes EMD-29754 (apical end of *C. parvum*), EMD-29755 (basal end of *C. parvum*), and EMD-29753 (apical end of *T. gondii*). Subtomogram averages of the PCRs are available in the EMDB under the accession codes EMD-29784 (*C. parvum* PCRs), EMD-29791 (refined upper PCR), and EMD-29801 (refined lower PCR). Subtomogram averages of IMCSF available in the EMDB under accession codes EMD-29808 (IMCSF top view), EMD-29809 (IMCSF sawtooth view), and EMD-29810 (IMCSF side view). The subtomogram average of the basal IMC pore is available in the EMDB under the accession code EMD-29835 (basal IMC pore). Subtomogram averages of the PCRs are available in the EMDB under accession codes EMD-29832 (wildtype *T. gondii* PCRs), EMD-29826 (refined upper PCR), and EMD-29827 (refined lower PCR). Subtomogram averages are available in the EMDB under the accession codes EMD-29838 (FRM1-iKD *T. gondii* PCRs), EMD-29839 (refined upper PCR), and EMD-29840 (refined lower PCR). Source data are provided with this paper.

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

## Acknowledgements

We thank Dr. Stefan Steimle for technical assistance with the Titan Krios G3i cryogenic electron microscope, the Singh Center for Nanotechnology and the Beckman Center for Cryo-Electron Microscopy at the University of Pennsylvania for hosting and supporting the use of the Titan Krios; we thank Liam Theveny for his assistance with generating the initial subtomogram average of the basal pore; we thank Dr. Benjamin Barad and Dr. Danielle Grotjahn at The Scripps Research Institute for their help with analyses related to F-actin organization. This work was supported in part by a David and Lucile Packard Fellowship for Science and Engineering (2019–69645), Burroughs Wellcome Fund Investigators in the Pathogenesis of Infectious Disease Program (1022785), and a Pennsylvania Department of Health FY19 Health Research Formula Fund to Y.-W.C.; a Martin and Pamela Winter Infectious Disease Fellowship to M.M.; the Mary L. and Matthew S. Santirocco College Alumni Society Undergraduate Research Grant to W.D.C.; an EMBO fellowship (ALTF 58–2018) to A.G.; the Swiss National Foundation (grant Nos. 310030_185325) to D.S.-F.; and grants R01AI112427 and R01AI127798 from the National Institutes of Health to B.S.

## Author contributions

S.K.M., M.M., A.G., B.S., and Y.-W.C. conceptualized and designed the experiments. A.G. performed all parasite cultures and preparation. S.K.M. and A.G. prepared frozen grids and performed cryo-ET for *C. parvum* and wildtype *T. gondii*. M.M. and A.G. prepared frozen grids and performed cryo-ET for FRM1-iKD *T. gondii*. W.D.C. established an automated data-processing pipeline for on-the-fly tomogram reconstruction and also provided additional computational support during data collection, processing, and management. M.M., S.K.M., C.P.T., and W.D.C. analyzed the tomograms. M.M. performed subtomogram averaging and analyses pertaining to apical F-actin, IMCSFs, PCRs, and basal IMC pores. S.C. assisted with subtomogram averaging of the IMCSFs. A.G. assisted with subtomogram averaging of the PCR from FRM1-iKD *T. gondii*. W.D.C., M.M., and S.K.M. performed the analysis of basal end F-actin. D.S.-F. provided the FRM1-iKD *T. gondii* strain. M.M. and S.K.M. prepared the manuscript, with critical inputs and revisions from all authors.

## Competing interests

The authors declare no competing interests.
