## [Peer Review File · Nature Communications]

Origin and arrangement of actin filaments for gliding motility in apicomplexan parasites revealed by cryo-electron tomographyReviewers' Comments:

Reviewer #1:

Remarks to the Author:

Overall: This manuscript elegantly describes many new ultrastructural features of the unusual machinery used for gliding motility in the apicomplexan parasites using cryo-electron tomography and subtomogram averaging to enhance resolution of regular structures. This new evidence unifies prior observations in the areas of motility, actin biochemistry and morphology and provides important new structural information concerning actin and other filaments in *C. parvum*. This manuscript is data dense yet clearly written with high quality figures and careful quantification. Ultimately, I have very minor suggested changes to it and I think it will be an extremely important contribution to our understanding of the cytoskeleton and motility in apicomplexans.

Comments for clarification:

1. A FRM1 induced knockdown in *T. gondii* was used to extend analysis of the conoid associated formin. I assume (but did not see explicitly stated) that F-actin filaments are entirely absent in the iKD samples? This might be nice to state if true. I also would assume that the IMCSFs are unchanged in appearance in the iKD? Adding both details to the last paragraph in the results section would be helpful.
2. Do you think that the IMCSFs are alveolins or a separate system of filaments? It's tempting to conclude that IMCSFs are alveolins tacked to the IMC by GAPM proteins. The IMCSFs nicely fit the associated filaments that maintain the regular array of IMPs in the IMC. Given the length of this manuscript, I am also fine with this not being added to the discussion at the discretion of the authors.

Corrections:

Line 187: "showed regular spaced between them" spaced should be spacing

Line 205-7: "Overall, our data suggests that IMCSFs guide F-actin along a helical path down the cell body of *C. parvum* to promote helical gliding motility while MyoA is likely organized longitudinally between IMCSFs." This seems much more like a sentence that belongs in the discussion section.

Reviewer #2:

Remarks to the Author:

This manuscript by Martinez et al is to characterize the organization of F-actin in extracellular *Cryptosporidium* parasites using electron cryo-electron tomography. In Apicomplexan parasites, as in most eukaryotes, actin is an essential protein that drives a wide range of cellular processes including parasite invasion and motility, apicoplast inheritance and vesicle transport. The biophysical properties of Apicomplexan actin and its organization in the cell remains controversial and so the experiments presented here have the potential to provide new insight into the organization of actin during motility. The data presented in this current manuscript provides new insights into the positioning of FRM1, which nucleates actin filaments, into the upper PCR and provides new insights into IMC "filaments" that likely guide actin during apical-basal translocation in the peripheral IMC space. However, a number of other conclusions are not well supported by the data, as presented in the manuscript and requires further explanation and analysis.

Major concerns:

1. Measured actin lengths are a function of focal plane/contrast in the sample and not indicative of the actual lengths of F-actin in the cell. Figure 1D, H, G/Line 101. From the graph in Fig. 1G, actin lengths had an average value of approx. 100nm. However, the ends of the identified filaments in Fig. 1D and H (highlighted in purple), coincides with a decrease in image contrast suggesting that measured

lengths are not truly reflective of the length of actin filaments in the cell but influenced significantly by the slice of the tomograph. To this point, the lengths of actin filaments shown in Fig. 4D appear to have a mean length of 300nm based on my crude estimation. The lengths of actin found in the IMC space should be quantified. The biophysical properties of Apicomplexan actin remain controversial, and it is still debated if these actin isoforms can form filaments longer than 200nm. Thus, an accurate measurement of actin in the cell is a crucial piece of this puzzle.

2. Quantification of F-actin/cell: In Figure 2D, the authors quantified the number of actin filaments in 20 cells. None of these cells contained zero actin filaments. The authors, however, analyzed a total of 119 untreated *C. parvum* tomographs. Authors should state how many of the images analyzed did not contain actin filaments. To test if the analyze pipelines are robust, and accurately identify actin filaments an analysis on cytochalasin D treated *C. parvum* should be performed. If the analyze pipelines are accurately identifying actin filaments, then there should be a significant decrease in the number of filaments identified after CytoD treatment. To this point, in Figure 5G, it is unclear why some signal is annotated as actin. It looks remarkably similar to the background signal (see point 4 below). CytoD treatment would help clarify these annotations.

3. The data in Figure 3B which identifies a putative F-actin connector is unconvincing. From the images shown the density assigned as this linker is not visible. Authors should show the raw data without the pseudo-coloring so readers can evaluate the raw data, and quantification on the number of times this structure was identified should be performed.

4. Similar to point 3, the data for the existence of the F-actin-pore connector in Figure 5G is also unconvincing. Authors should state how many images these linkers were observed and show the raw data in the absence of pseudocoloring. It is also unclear how the actin filaments in 5G themselves were assigned given the noise in the images. The "background" grey signal that has not been annotated as actin that looks remarkably similar to the signal under the annotated F-actin filaments. Authors should show the data without pseudo-coloring so raw data is visible.

5. Aspects of model outlined in the conclusion (starting at line 346) are not supported by the data, particularly the sentence "MyoH motors tethered to the conoid walk on the F-actin to power conoid extrusion". From previously published data, it is clear that MyoH is playing SOME role in conoid extension but it is unclear, if or how MyoH provides the force for conoid protrusion. My concern about this model is the following: The majority of formin proteins bind to the growing + end of actin filaments, so we could assume that the + end of actin is anchored into the PCR and the - end is pointing away from the conoid. This orientation is consistent with MyoA, a + end directed myosin motor, providing the force for invasion. So, if we assume that the -end of actin is tether to IMC collar (although in my view the data to support this interaction is weak, see point 4 above), and that MyoH is a + end directed motor since it does not contain the unique inserts that are found in Myo6 (which confer reverse directionality) then MyoH would translocate filaments along the conoid moving towards the basal end. If the - ends of the filaments are tethered to the IMC collar, MyoH translocation would result in actin filament buckling. Actin is a flexible polymer and force cannot be generated in this way, just like we could not generate force by pushing on a rope. The authors need to articulate clearly their hypothesis for how MyoH motors can provide the force for conoid protrusion or remove this statement from the paper.

6. Introduction: Line 52 and 102. "F-actin used for gliding is very short and unstable" and "short median length is.....characteristic of apicomplexan F-actin" The basis for this statement comes from in vitro data with recombinant actin and is an oversimplification of the literature. There isn't strong in vivo data demonstrating the length of actin in the Apicomplexa and there is conflicting data about the filament length and critical concentration of actin in vitro. Authors should provide a more comprehensive and in-depth discussion of the conflicting data in the literature including but not limited to the following papers: Lu et al 2020 PMID: 31548388; Kumpula et al 2019 PMID: 31199804; Kumpula et al 2017 PMID 28939886.

Reviewer #3:

Remarks to the Author:

In this study the authors used cryo electron tomography (cryo-ET) to investigate F-actin organization within motile *C. parvum* sporozoites and *T. gondii* tachyzoites. The obtained ultrastructural and structural data support a model in which F-actin nucleation occurs at the apical tip of these parasites and it is linked to the conoid extrusion.

The work is well structured and presented clearly, with beautiful illustrations that guide the reader effortlessly throughout the experiments. Set aside the FRM1 localisation (see my comments below), the conclusions are always phrased appropriately and supported by the presented data. This in situ structural study offers novel and exciting insights in apicomplexan motility, I recommend it for publication after few minor edits.

Main comments:

Line 65: remove the word "precisely". The data shown does not support a precise localization of FRM1. However, it does provide indications about FRM1 position with respect to the PCR and its possible role as F-actin nucleator at the upper pre-conoidal ring.

Lines 270-271: The obtained wild type and FRM1-KD subtomogram averages do not support this statement. A convincing fit for the small TPR domain requires secondary structure elements. While it is good to show the difference map and suggest a plausible explanation, as reported in lines 267-269, I suggest removing this sentence (lines 270-271) and the fit shown for the TPR domain in Figure 6H unless more convincing structural evidence is provided.

Minor comments:

Lines 120-123: from the presented data, it is convincing that the F-actin filaments end in proximity of the upper PCR, often close to what the authors defined as a "protruding density". However, the observed variability of the F-actin filaments ends with respect to the PCR could be explained by the deformation that the cells experience in the sample preparation step. It is known that apicomplexan cells, although they can be prepared by standard plunge-freezing for cryo-ET, do suffer from mechanical compression during the blotting phase that results in a considerable deformation of the cell before vitrification. From the supplementary video 2 it is evident how the cell is deformed in the process, with the PCR architecture being far from a circular ring. Could it be that this mechanical deformation results in an inconsistent displacement of the filaments from the PCR? The authors mention something similar in the Material and Methods section "Quantifications, statistics, and reproducibility". I suggest briefly elaborating on this problem in the main text, pointing to the Material and Methods section for further details.

Point-by-point responses to the reviewers' comments

We would like to sincerely thank all 3 reviewers for their time, constructive comments to help improve the manuscript, and their enthusiasm for publication. Below, we provide our point-by-point response that carefully addresses each of the comments.

Reviewer #1 (Remarks to the Author):

Overall: This manuscript elegantly describes many new ultrastructural features of the unusual machinery used for gliding motility in the apicomplexan parasites using cryo-electron tomography and subtomogram averaging to enhance resolution of regular structures. This new evidence unifies prior observations in the areas of motility, actin biochemistry and morphology and provides important new structural information concerning actin and other filaments in *C. parvum*. This manuscript is data dense yet clearly written with high quality figures and careful quantification. Ultimately, I have very minor suggested changes to it and I think it will be an extremely important contribution to our understanding of the cytoskeleton and motility in apicomplexans.

We greatly appreciate the reviewer's positive feedback.

Comments for clarification:

1. A FRM1 induced knockdown in *T. gondii* was used to extend analysis of the conoid associated formin. I assume (but did not see explicitly stated) that F-actin filaments are entirely absent in the iKD samples? This might be nice to state if true.

Thank you for pointing this out. Indeed, we did not see F-actin in the iKD samples. We have now added the sentence –

Line 273: "*In tomograms of TgFRM1-iKD, no F-actin was observed across all 28 cells imaged.*"

I also would assume that the IMCSFs are unchanged in appearance in the iKD? Adding both details to the last paragraph in the results section would be helpful.

We did not observe any IMCSFs in *T. gondii*, unlike in *C. parvum*. It is unknown if an inducible FRM1 knockdown would impact IMCSF structure in *C. parvum* because the mutant is currently unavailable. To clarify we added the following sentence to the manuscript –

Line 389: “Interestingly, IMCSFs seem to be a species-specific adaption in *C. parvum* as *T. gondii* tachyzoites do not show any similar features. It is therefore possible that the long SPMTs in certain species including *T. gondii* are sufficient to guide F-actin in helical paths down the parasite body.”

2. Do you think that the IMCSFs are alveolins or a separate system of filaments? It's tempting to conclude that IMCSFs are alveolins tacked to the IMC by GAPM proteins. The IMCSFs nicely fit the associated filaments that maintain the regular array of IMPs in the IMC. Given the length of this manuscript, I am also fine with this not being added to the discussion at the discretion of the authors.

We did not observe any IMCSFs or IMCSF-like filaments in the *T. gondii* pellicular space, unlike in *C. parvum*. The spacing between IMCSFs in *C. parvum* is indeed reminiscent of the ~30 nm spacing between the columns of IMPs previously described in *T. gondii* (PMID: 9010782). Moreover, six alveolin encoding genes are known to be conserved in the *C. parvum* genome (PMID 18359944). However, we are hesitant to conjecture that the IMCSFs could be alveolins. This is because the IMCSFs are found on the IMC surface in the pellicular space while the alveolins are predicted to be on the cytoplasmic face of the IMC mediating interactions with the subpellicular microtubules. Moreover, the intercalated architecture of IMCSFs and subpellicular microtubules on either side of the IMC seen in our data suggests that the alveolins are likely arranged in the space between neighboring IMCSFs. Nonetheless, we cannot rule out the possibility that alveolins help organize the IMCSFs. Due to the amount of speculation required to discuss these possibilities with sufficient clarity, we decided to not include this in the discussion of this manuscript. Nevertheless, we are grateful for the reviewer's stimulating suggestion.

Corrections:

Line 187: “showed regular spaced between them” spaced should be spacing

Thank you. We have changed “spaced” to “spacing” (new line 171).

Line 205-7: “Overall, our data suggests that IMCSFs guide F-actin along a helical path down the cell body of *C. parvum* to promote helical gliding motility while MyoA is likely organized longitudinally between IMCSFs.” This seems much more like a sentence that belongs in the discussion section.

Thank you for the comment. We have now removed “*while MyoA is likely organized longitudinally between IMCSFs*” from the Results section (line 188), as this concept is more appropriately discussed in the discussion section (lines 380-382).

Reviewer #2 (Remarks to the Author):

This manuscript by Martinez et al is to characterize the organization of F-actin in extracellular *Cryptosporidium* parasites using electron cryo-electron tomography. In Apicomplexan parasites, as in most eukaryotes, actin is an essential protein that drives a wide range of cellular processes including parasite invasion and motility, apicoplast inheritance and vesicle transport. The biophysical properties of Apicomplexan actin and its organization in the cell remains controversial and so the experiments presented here have the potential to provide new insight into the organization of actin during during motility. The data presented in this current manuscript provides new insights into the positioning of FRM1, which nucleates actin filaments, into the upper PCR and provides new insights into IMC “filaments” that likely guide actin during apical-basal translocation in the peripheral IMC space. However, a number of other conclusions are not well supported by the data, as presented in the manuscript and requires further explanation and analysis.

We thank the reviewer very much for providing a constructive critique to help us strengthen this manuscript.

Major concerns:

1. Measured actin lengths are a function of focal plane/contrast in the sample and not indicative of the actual lengths of F-actin in the cell. Figure 1D, H, G/Line 101. From the graph in Fig. 1G, actin lengths had an average value of approx. 100nm. However, the ends of the identified filaments in Fig. 1D and H (highlighted in purple), coincides with a decrease in image contrast suggesting that measured lengths are not truly reflective of the length of actin filaments in the cell but influenced significantly by the slice of the tomograph. To this point, the lengths of actin filaments shown in Fig. 4D appear to have a mean length of 300nm based on my crude estimation. The lengths of actin found in the IMC space should be quantified. The biophysical properties of Apicomplexan actin remain controversial, and it is still debated if these actin isoforms can form filaments longer than 200nm. Thus, an accurate measurement of actin in the cell is a crucial piece of this puzzle.

We thank the reviewer for raising this concern. We are sorry for causing any misunderstanding of how we measured F-actin length. Indeed, if one measured F-actin lengths from only a single 2-D image slice through a 3-D tomogram, the result will not be indicative of the actual lengths of F-actin in the cell. Therefore, we performed length measurements for this study by carefully tracing each F-actin filament in 3 dimensions through multiple slices/angles in the tomograms. We would like to note that the actin filaments highlighted in Fig. 1D, H, and Fig. 4D are

shown for display purposes to indicate the F-actin location in these 2-D slices cutting through 3-D tomograms. Measurements were performed on the 3-D tracing in tomograms. As for the contrast issue, we chose tomograms with the best contrast for quantification; these tomograms offered sufficient contrast to discern F-actin filament ends in 3-D. We have made the following modifications to the Methods –

Line 498: *“IMOD models were generated with filament contours, where each filament was manually and carefully traced in 3-D through multiple slices/angles in the tomograms to help accurately estimate filament lengths. Our tomograms offered sufficient contrast to discern filament ends in 3-D. Representative 2-D slices cutting through the 3-D tomograms (for example, figure panels Fig. 1D, H, and Fig. 4D that show annotated filaments) were chosen for display purposes.”*

2. Quantification of F-actin/cell: In Figure 2D, the authors quantified the number of actin filaments in 20 cells. None of these cells contained zero actin filaments. The authors, however, analyzed a total of 119 untreated *C. parvum* tomographs. Authors should state how many of the images analyzed did not contain actin filaments.

To clarify, we reconstructed tomograms from a total of 119 untreated *C. parvum* cells (from 4 multi-day sessions) but for each detailed analysis, we quantified from only a subset of these tomograms based on best contrast but otherwise random. For F-actin quantification in Figure 2D, which was to understand the variability in F-actin number from cell-to-cell, we similarly analyzed 20 cells and did not find any cell that had zero filaments. This was not surprising since we very frequently found F-actin in the 119 *C. parvum* cells by cursory examination. However, we cannot rule out the possibility that some of the cells lack apical F-actin altogether. The manuscript is now modified as follows –

Line 474: *“We obtained a total of 228 *C. parvum* tomograms (119 untreated, 109 jasplakinolide-treated) and 128 *T. gondii* tomograms (100 wildtype, 28 FRM1-iKD). For each sample, multiple frozen grids were imaged over several sessions, each spanning multiple days. Each of the following quantifications is from a subset of these tomograms, chosen based on best contrast but otherwise random.”*

Line 956: *“2-D tomogram slices showing F-actin ends near the upper PCR, and a scatter plot of the number of PCR-associated F-actin observed per cell (n = 20 cells).”*

To test if the analyze pipelines are robust, and accurately identify actin filaments an analysis on cytochalasin D treated *C. parvum* should be performed. If the analyze pipelines are accurately identifying actin filaments, then there should be a significant decrease in the number of filaments identified after CytoD treatment. To this point, in Figure 5G, it is unclear why some signal is annotated as actin. It looks remarkably similar to the background signal (see point 4 below). CytoD treatment would help clarify these annotations.

To clarify, we performed the identification and tracing of apical F-actin manually, instead of using an automated pipeline. We had two people (M Martinez and S Mageswaran) independently examine the results to ensure accurate identification of actin filaments. We therefore believe that we should have been able to spot cells with no apical actin filaments in the *C. parvum* dataset, if any. As further proof for the robustness of our F-actin annotation, we are able to corroborate that (1) jasplakinolide-treated *C. parvum* cells showed more abundant actin filaments; and (2) *T. gondii* after induced knockdown of Formin1 showed no actin filament. For these reasons, we respectfully deem CytoD treatment unnecessary.

With regards to Figure 5G, we agree that some tomograms contain higher background noise (due to sample thickness), rendering it more difficult to precisely annotate actin filament densities. However, we would like to point out that the representative image in Fig. 5G (a 2-D slice through a 3-D tomogram) is for illustration purposes only to show F-actin along with other pertinent features. For filament annotation/tracing, we accessed the original 3-D tomogram using multiple slices and angles. While annotating a single 2-D slice can result in false positives and false negatives, using the entire tomogram to examine if the F-actin density is continuous in the neighboring slices in 3-D can help to significantly reduce such instances. We have made the following clarification in the manuscript –

Lines 498-508: *“IMOD models were generated with filament contours, where each filament was manually and carefully traced in 3-D through multiple slices/angles in the tomograms to help accurately estimate filament lengths. Our tomograms offered sufficient contrast to discern filament ends in 3-D. Representative 2-D slices cutting through the 3-D tomograms (for example, figure panels Fig. 1D, H, and Fig. 4D that show annotated filaments) were chosen for display purposes. It is noteworthy that while annotating single 2-D slices from tomograms can result in false positives and false negatives, our use of entire tomograms to examine filament continuity into neighboring slices helped to significantly reduce such*

instances. To further ensure accurate identification of actin filaments, two people (M.M. and S.K.M) independently examined the results.”

3. The data in Figure 3B which identifies a putative F-actin connector is unconvincing. From the images shown the density assigned as this linker is not visible. Authors should show the raw data without the pseudo-coloring so readers can evaluate the raw data, and quantification on the number of times this structure was identified should be performed.

Thank you for the comment. We agree and apologize that the connector density is difficult to observe with color overlay. We had included in the original manuscript the corresponding raw 2-D slice in Supp. Fig. 11 to help readers carefully assess the annotation for themselves but failed to reference it properly. We have now included the following reference within Fig. 3 –

Line 989: *“Unannotated images (without color overlays) for panels A and B are included in Supp. Fig. 11.”*

Although quantification of the observation of this density would be ideal, we acknowledge that we are working at the limits of detection to reliably annotate this small feature. Therefore, we have taken another approach to evaluate the tethering of F-actin to the IMC collar – we have measured the shortest distance between the F-actin and the IMC collar whenever we see an actin filament enters the cytosolic space along with IMC collar properly resolved (n=22). This new result is reported in new Fig. 3D –

Line 149: *“Consistent with the presence of such tethers, we observed the closest distance between cytoplasmic F-actin and the IMC collar to be fairly conserved (~12 nm) (Fig. 3D).”*

Line 986: *“(D) Violin, box and beeswarm plots for the closest distance between cytoplasmic F-actin and the IMC collar (n=22).”*

We have also added more examples of this putative connector in the new Supp. Fig. 2C.

Line 1075: *“(C) A gallery of 2-D slices from C. parvum sporozoite tomograms showing putative connector densities between cytoplasmic F-actin and IMC collar.”*

Despite these newly included results, we have carefully worded our observations and discussions on these putative connectors –

Line 147: “*Interestingly, PCR-associated filaments that entered the cytoplasm were guided over the conoid and were often seen associating with the IMC collar via putative connectors (Fig. 3B – lower panel and Supp. Fig. 2B, C).*”

Line 332: “*The cytoplasmic F-actin is tentatively anchored at the IMC collar through putative connectors*”

Line 981: “*a putative density is observed linking cytoplasmic F-actin to the IMC collar*”

Line 1074: “*interactions with the IMC collar via putative connector densities.*”

4. Similar to point 3, the data for the existence of the F-actin-pore connector in Figure 5G is also unconvincing. Authors should state how many images these linkers were observed and show the raw data in the absence of pseudocoloring. It is also unclear how the actin filaments in 5G themselves were assigned given the noise in the images. The “background” grey signal that has not been annotated as actin that looks remarkably similar to the signal under the annotated F-actin filaments. Authors should show the data without pseudo-coloring so raw data is visible.

As mentioned in our responses to the previous points, we always accessed the original 3-D tomograms using multiple slices and angles to annotate structural features and we present the 2-D slices in figures only for display purposes. We had originally included unannotated images corresponding to Fig. 5G in Supp. Fig. 12 to help readers assess the annotations for themselves. However, we realize that we had not referenced them properly, which we have now rectified –

Line 1037: “*Unannotated images (without color overlays) for panels G and H are included in Supp. Fig. 12.*”

Moreover, we have done the same for other annotated figures as we have done for Fig. 3 and Fig. 5 –

Line 940: “*Unannotated images (without color overlays) for all relevant figure panels are included in Supp. Fig. 10.*”

Line 971: “*Unannotated images (without color overlays) for panels B, C and E are included in Supp. Fig. 11.*”

Line 1010: “*Unannotated images (without color overlays) for panel G are included in Supp. Fig. 12.*”

Line 1064: “*Unannotated images (without color overlays) for panel F are included in Supp. Fig. 12.*”

Although the pore-actin connectors are tricky to observe consistently in individual 3-D tomograms, our confidence in the presence of this feature is boosted by its appearance in the subtomogram average (Fig. 5H and Supp. Fig. 12), which is now highlighted.

Nonetheless, we have now taken additional care in discussing this feature –

Line 240: “*Moreover, putative connecting densities were seen between the pores and nearby F-actin (Fig. 5G) that was retained in the average structure as well (Fig. 5H).*”

Line 401: “*We observed pores embedded within the IMC that are specifically positioned at the basal F-actin cap, occasionally showing tentative associations with F-actin.*”

Line 1031: “*(G) 2-D slices from a basal end tomogram displaying side views of an IMC pore tentatively associating with F-actin in the pellicular space.*”

Line 1036: “*the densities annotated with dashed orange outlines likely represent putative connectors between the pore and F-actin.*”

5. Aspects of model outlined in the conclusion (starting at line 346) are not supported by the data, particularly the sentence “MyoH motors tethered to the conoid walk on the F-actin to power conoid extrusion”. From previously published data, it is clear that MyoH is playing SOME role in conoid extension but it is unclear, if or how MyoH provides the force for conoid protrusion. My concern about this model is the following: The majority of formin proteins bind to the growing + end of actin filaments, so we could assume that the + end of actin is anchored into the PCR and the – end is pointing away from the conoid. This orientation is consistent with MyoA, a + end directed myosin motor, providing the force for invasion. So, if we assume that the -end of actin is tether to IMC collar (although in my view the data to support this interaction is weak, see point 4

above), and that MyoH is a + end directed motor since it does not contain the unique inserts that are found in Myo6 (which confer reverse directionality) then MyoH would translocate filaments along the conoid moving towards the basal end. If the – ends of the filaments are tethered to the IMC collar, MyoH translocation would result in actin filament buckling. Actin is a flexible polymer and force cannot be generated in this way, just like we could not generate force by pushing on a rope. The authors need to articulate clearly their hypothesis for how MyoH motors can provide the force for conoid protrusion or remove this statement from the paper.

We thank the reviewer for discussing the implications of our proposed model in detail. We take this opportunity to reiterate and substantiate our working model: MyoH mediates extrusion of the conoid using cytosolic F-actin, a process that subsequently gates F-actin into the pellicular space to drive MyoA-mediated motility.

- (1) FRM1 and conoid-associated MyoH are essential for conoid extrusion, suggesting a role for F-actin-MyoH interactions in this process (PMID 36109645).
- (2) Such conoid extrusion is essential for subsequent MyoA-mediated motility, likely involving the proper localization of F-actin into the pellicular space (PMIDs 30753127, 26760042, 36109645). Pellicular F-actin was previously predicted to drive motility and several observations from this study support this model. Consistent with conoid extrusion being a prerequisite for pellicular localization of F-actin, our tomograms of parasites with retracted conoid show abundant F-actin in the cytoplasm instead. Moreover, it is evident from our tomograms that a fully retracted conoid would position the PCR within/below the IMC collar. F-actin generated at the PCR in this conformation cannot efficiently enter the pellicular space.
- (3) Our observations of enriched cytosolic F-actin in the context of retracted conoid suggests that this pool of F-actin may facilitate conoid extrusion, leading to our model: MyoH (along with its tethered conoid) walks on these cytosolically localized F-actin tracks to extrude the conoid. We acknowledge the reviewer's concerns regarding how F-actin, being a soft polymer, might be incapable of offering a still track (anchored to the IMC collar) for the conoid to "push" forward. This prompted us to think more carefully and re-inspect our tomograms. It struck us that the tentative connector density is positioned in front of the retracted conoid (e.g., Fig. 3B (bottom), Supplementary Fig. 2B), which suggests that the conoid is "pulling" forward on F-actin that may be anchored at the IMC collar. Such pulling activity (as opposed to "pushing") is more feasible for actomyosin systems in terms of force generation and is compatible with the arrangement of the retracted

conoid. Additionally, such F-actin tethering at the IMC collar, along with potentially multiple MyoH copies binding to each actin filament, surface adhesins possibly immobilizing these filaments with respect to the gliding surface above the IMC collar, and other unknown F-actin binding proteins could together help stiffen the F-actin tracks. Furthermore, multiple F-actins functioning in concert could reduce the force on each filament. Together, these factors could facilitate F-actin to function as tracks for conoid extrusion, especially as the conoid begins to move past the IMC collar. A fully extruded conoid sits just above the IMC collar (e.g., Fig. 3A, 3B (top)) and would not need to further “push” down on the F-actin which may cause buckling of the filament. Our model requires and assumes the controlled elongation of actin filaments at the PCR, or their detachment from the PCR, as the conoid progressively extrudes. We imagine this model for conoid extrusion to share several aspects with that of gliding motility powered by MyoA. As for F-actin orientation, we agree with the reviewer’s model i.e. “+” ends oriented towards the PCR not just based on the predicted directionality of MyoH and MyoA but also based on FRM1 localization to the PCRs.

- (4) We understand that this force generation model for conoid extrusion is speculative and is beyond the scope of this study. Nonetheless, we felt the need to help organize several important and otherwise disparate observations related to conoid extrusion to serve as a primer for subsequent research in this direction. We have therefore taken utmost care while briefly discussing this model, clearly highlighting its speculative aspect as indicated below –

Line 328: “*Integrating published findings and our in situ structural observations, we propose the following mechanistic model, albeit speculative, for F-actin gating into the pellicular space (**Supp Fig. 8C**): Briefly, F-actin is nucleated at the PCRs, elongated, and channeled into the cytoplasm. The cytoplasmic F-actin is tentatively anchored at the IMC collar through putative connectors while FRM1 continues to elongate the F-actin at the PCRs in a regulated fashion. Concomitantly, MyoH motors tethered to the conoid walk on the F-actin to power conoid extrusion, a process that may be assisted by other factors such as the Glideosome Associated Connector protein (GAC). When the conoid extrudes, the PCRs are placed sufficiently above the IMC collar allowing the channeling of newly generated F-actin into the pellicular space to power motility (see **Supp. Fig. 8C** for more details).”*

Line 1146: “**(C)** *A speculative mechanistic model for coupling of conoid extrusion with F-actin polymerization and their pellicular delivery. First, when the conoid is retracted (left), FRM1-nucleated F-actin is channeled over the conoid and into the*

cytoplasm close to the IMC. There, the filaments are potentially anchored to the IMC collar. Using these filaments as “anchored tracks/cables”, the conoid-associated MyoH could potentially “pull” the conoid forward to extrude it. Tethering of these filaments at the IMC collar, binding of multiple MyoH molecules (and other as-yet-unknown F-actin binding factors) to each filament, and immobilization of these filaments with respect to the gliding surface through surface adhesins and Glideosome Associated Connector proteins (GAC) could together help to potentially stiffen these actin “tracks”. Furthermore, multiple filaments functioning in concert could also help to optimally distribute the force for extrusion. The filaments likely continue to elongate at the PCR in a regulated fashion (or detach from the PCR) until the conoid is fully extruded. Following conoid extrusion, newly nucleated F-actin can now be properly gated into the pellicular space (right).”

6. Introduction: Line 52 and 102. “F-actin used for gliding is very short and unstable” and “short median length is.....characteristic of apicomplexan F-actin” The basis for this statement comes from in vitro data with recombinant actin and is an oversimplification of the literature. There isn’t strong in vivo data demonstrating the length of actin in the Apicomplexa and there is conflicting data about the filament length and critical concentration of actin in vitro. Authors should provide a more comprehensive and in-depth discussion of the conflicting data in the literature including but not limited to the following papers: Lu et al 2020 PMID: 31548388; Kumpula et al 2019 PMID: 31199804; Kumpula et al 2017 PMID 28939886.

We agree to and thank the reviewer for this recommendation. We have now carefully discussed the existing literature about apicomplexan F-actin.

Introduction (Line 30): “Much of our understanding of the nature of apicomplexan F-actin related to gliding motility comes from in vitro studies, mainly from *Plasmodium* and *Toxoplasma*. Although there is an ongoing debate over the critical concentration and the ability to form long filaments, several studies indicate that apicomplexan F-actin is more unstable compared to its metazoan counterpart and likely forms short filaments in vivo, a property that has likely hindered experimentation and conceptual advancement.”

Results (Line 81): “These filaments exhibited a diameter of 7–10 nm and a median length of 125 nm (**Fig. 1G**), resembling short F-actin predicted in apicomplexan parasites in some previous studies.”

Discussion (Line 287): “*Biochemical and structural studies of Toxoplasma and Plasmodium actin suggest that their filaments are likely short and unstable in vivo, rendering them difficult to study.*”

Additionally, we have also taken efforts to properly present our findings on *C. parvum* F-actin in the context of apicomplexan F-actin in general. For example, we have noted the difference in the abundance of F-actin between *C. parvum* sporozoites and *T. gondii* tachyzoites in our tomograms and argue in favor of species-specific and life cycle stage-specific differences between the two –

Line 299): “*Interestingly, we observe a greater abundance of apical F-actin in C. parvum sporozoites when compared to T. gondii tachyzoites. This difference may reflect their different life cycle stages, isolation methods or species-specific dynamics (e.g., C. parvum sporozoites glide at higher speeds).*”

Reviewer #3 (Remarks to the Author):

In this study the authors used cryo electron tomography (cryo-ET) to investigate F-actin organization within motile *C. parvum* sporozoites and *T. gondii* tachyzoites. The obtained ultrastructural and structural data support a model in which F-actin nucleation occurs at the apical tip of these parasites and it is linked to the conoid extrusion.

The work is well structured and presented clearly, with beautiful illustrations that guide the reader effortlessly throughout the experiments. Set aside the FRM1 localisation (see my comments below), the conclusions are always phrased appropriately and supported by the presented data. This in situ structural study offers novel and exciting insights in apicomplexan motility, I recommend it for publication after few minor edits.

We thank the reviewer very much for the overall positive feedback on the manuscript.

Main comments:

Line 65: remove the word “precisely”. The data shown does not support a precise localization of FRM1. However, it does provide indications about FRM1 position with respect to the PCR and its possible role as F-actin nucleator at the upper pre-conoidal ring.

We have now removed the word “precisely”.

Lines 270-271: The obtained wild type and FRM1-KD subtomogram averages do not support this statement. A convincing fit for the small TPR domain requires secondary structure elements. While it is good to show the difference map and suggest a plausible explanation, as reported in lines 267-269, I suggest removing this sentence (lines 270-271) and the fit shown for the TPR domain in Figure 6H unless more convincing structural evidence is provided.

Thank you for the comment. We have now removed the line – “*Intriguingly, the predicted structure of the TPR domain fits well into this region (Fig. 6H)*” from the results section. We have also moved Fig. 6H to Supp. Fig. 9D and only talk about this possibility in the discussion section –

Line 313: “*Our findings support a model in which the FRM1 N-terminal region (including the TPR domain) anchors the protein at the protruding density of the PCR while the C-terminal FH2 domain flexibly reaches out to nucleate F-actin (Fig. 6H). Consistent with this idea, the predicted structure of the TPR domain*

seems to fit well into one of the missing densities in the subtomogram average of TgFRM1-iKD PCR that constitutes a portion of the protruding density (Supp. Fig. 9D)."

Line 1175: "**(D)** AlphaFold2 prediction of the TgFRM1 TPR domain structure, fit into one of the missing densities of the FRM1-iKD PCR at the protruding density (right), zoomed in on the black box in the left panel."

Minor comments:

Lines 120-123: from the presented data, it is convincing that the F-actin filaments end in proximity of the upper PCR, often close to what the authors defined as a "protruding density". However, the observed variability of the F-actin filaments ends with respect to the PCR could be explained by the deformation that the cells experience in the sample preparation step. It is known that apicomplexan cells, although they can be prepared by standard plunge-freezing for cryo-ET, do suffer from mechanical compression during the blotting phase that results in a considerable deformation of the cell before vitrification. From the supplementary video 2 it is evident how the cell is deformed in the process, with the PCR architecture being far from a circular ring. Could it be that this mechanical deformation results in an inconsistent displacement of the filaments from the PCR? The authors mention something similar in the Material and Methods section "Quantifications, statistics, and reproducibility". I suggest briefly elaborating on this problem in the main text, pointing to the Material and Methods section for further details.

We agree with the reviewer on this point. The majority of parasites do suffer from flattening due to the blotting during sample plunge-freezing, and this artifact is clearly observed in Supplementary Video 2 as indicated in our Material and Methods section "Quantifications, statistics, and reproducibility". In more detail, we have observed this flattening effect to be quite homogenous along the blotting direction, preserving the organization of several apical structures relative to each other (PMID 34404783 and 35817892). As for F-actin organization at the PCR, we expect the F-actin tips to be displaced, if any, along the blotting direction together with their associated PCR subunits, possibly producing minor perturbations in their relative positioning. We have therefore added the following sentence to the main text –

Line 125: "*However, we cannot rule out the possibility that this variability could be partly attributed to the mechanical compression of the parasite, an artifact of cryo-ET sample preparation by plunge freezing (see Methods - Quantifications, statistics, and reproducibility)*".

In the Methods, we have made some modifications and included additional discussions on the compression artifact –

Line 480: *“We also note that parasites flattened on the grid. However, this flattening, which was predominantly in the blotting direction and therefore probably caused by the blotting force, was mostly homogeneous in this direction and did not adversely affect the shape and organization of various structures and organelles, except for some flattening of the PCRs, APR and the conoid. Flattening could have added subtle variations to the relative positions of features, but their organizational patterns were evident despite the presence of such potential effects. Importantly, we do not expect it to adversely affect the positioning of F-actin tips with respect to the PCRs, causing only subtle variations, if any, along the blotting direction.”*

Reviewers' Comments:

Reviewer #1:

Remarks to the Author:

I am happy with the revisions made to the MS in response to my questions and the points raised by the other reviewers.

Reviewer #2:

Remarks to the Author:

The authors have adequately addressed the concerns raised in the previous review

Reviewer #3:

Remarks to the Author:

All reviewers comments were carefully addressed by the authors. I don't have any further concern and I recommend the manuscript for publication.